# Trans-Golgi protein TVP23B regulates host-microbe interactions via Paneth cell homeostasis and Goblet cell glycosylation

Ran Song [1], William McAlpine[1], Aaron M. Fond [1,2], Evan Nair-Gill[1], Jin Huk Choi [1], Elisabeth E. L. Nyström[3], Liisa Arike [4], Sydney Field [2], Xiaohong Li[1], Jeffrey A. SoRelle [1,5], James J. Moresco[1], Eva Marie Y. Moresco [1], John R. Yates 3rd [6], Parastoo Azadi[7], Josephine Ni[2], George M. H. Birchenough [4], Bruce Beutler [1] & Emre E. Turer [1,2] ✉

A key feature in intestinal immunity is the dynamic intestinal barrier, which separates the host from resident and pathogenic microbiota through a mucus gel impregnated with antimicrobial peptides. Using a forward genetic screen, we have found a mutation in *Tvp23b*, which conferred susceptibility to chemically induced and infectious colitis. Trans-Golgi apparatus membrane protein TVP23 homolog B (TVP23B) is a transmembrane protein conserved from yeast to humans. We found that TVP23B controls the homeostasis of Paneth cells and function of goblet cells, leading to a decrease in antimicrobial peptides and more penetrable mucus layer. TVP23B binds with another Golgi protein, YIPF6, which is similarly critical for intestinal homeostasis. The Golgi proteomes of YIPF6 and TVP23B-deficient colonocytes have a common deficiency of several critical glycosylation enzymes. TVP23B is necessary for the formation of the sterile mucin layer of the intestine and its absence disturbs the balance of host and microbe in vivo.

Intestinal homeostasis results from a mutualistic relationship between the mucosal epithelium, immune system, as well as the host microbiome. Failure to maintain intestinal homeostasis can result in an increased susceptibility to gastrointestinal infections and chronic inflammation such as seen in ulcerative colitis and Crohn's disease[1–3]. At the tissue level, intestinal homeostasis depends on many factors including controlled immune cell activation, mucus secretion, epithelial cell integrity, and cell proliferation[4,5]. The secreted mucus layer is a critical barrier preventing translocation of commensal and pathogenic microbes and is comprised of a heavily glycosylated and antimicrobial peptide impregnated layer of mucus[1]. Despite its

fundamental role in regulating host/microbe interactions, the mechanisms through which Paneth and goblet cells maintain this sterile layer remain unclear.

To identify proteins with non-redundant function in homeostasis of the gastrointestinal tract, we used a forward genetic screen of N-ethyl-N-nitrosourea (ENU) mutagenized mice and determined mutations leading to hypersensitivity to dextran sodium sulfate (DSS)[5,6]. Through this screen, we found that loss of *Tvp23b* leads to hypersensitivity to DSS-induced colitis. Trans-Golgi apparatus membrane protein TVP23 homolog B (TVP23B) is a 22-kilodalton transmembrane protein represented in mammals, yeast and plants. The yeast homolog,

[1]Center for the Genetics of Host Defense, University of Texas Southwestern Medical Center, Dallas, TX 75390-8505, USA. [2]Department of Internal Medicine, Division of Gastroenterology, University of Texas Southwestern Medical Center, Dallas, TX 75390-8505, USA. [3]Institute of Biochemistry, University of Kiel, 24118 Kiel, Schleswig-Holstein, Germany. [4]The Wallenberg Centre for Molecular & Translational Medicine, Department of Medical Biochemistry, Institute of Biomedicine, University of Gothenburg, 40530 Gothenburg, Sweden. [5]Department of Pathology, University of Texas Southwestern Medical Center, Dallas, TX 75390-8505, USA. [6]Department of Molecular Medicine, The Scripps Research Institute, 10550 North Torrey Pines Road, La Jolla, CA 92037, USA. [7]Complex Carbohydrate Research Center, University of Georgia, Athens, GA 30602, USA. ✉e-mail: emre.turer@utsouthwestern.edu

TVP23, was originally found by proteomic analysis of Golgi compartments and was able to regulate vesicular traffic to and from this organelle[7,8]. The Arabidopsis homolog, Echidna, is required for secretory trafficking and loss of function leads to abnormal trans-Golgi protein localization as well as cellular elongation[9]. However, the function of the mammalian protein, TVP23B, and its role in the maintenance of intestinal homeostasis has not previously been described.

In this study, we report that TVP23B is necessary for the maintenance of intestinal homeostasis via the production of the sterile intestinal mucus layer. By regulating the proteome of the Golgi and localization of glycosyltransferases, TVP23B deficiency leads to abnormal glycosylation and loss of antimicrobial peptides and mucus. Mice lacking TVP23B showed loss of host−microbe segregation and increased susceptibility to infectious colitis. Thus, we show that TVP23B is an essential component of intestinal barrier function in vivo.

## Results

### TVP23B deficiency leads to colitis susceptibility

Random mutations were generated using ENU and a previously described inbreeding scheme was employed to evaluate mice in the heterozygous and homozygous mutant state[5,6,10]. We subjected 55,867 third generation (G3) mice from 2039 pedigrees to DSS in their drinking water, and body weights of the mice were recorded after 7 days of DSS treatment[5,11,12] (Fig. S1a). Among the 55,867 G3 mice tested, a total of 306 mice (0.5% of total) displayed weight loss of 20% by day 7 of DSS treatment (Fig. S1b). A weight loss phenotype was observed in mice from pedigree R4840 and the DSS-induced weight loss phenotype was mapped using an autosomal semidominant model of inheritance. The phenotype was designated *Chipotle*. The *Chipotle* phenotype mapped to a putative donor splice site mutation (a G > T transversion of the third nucleotide of intron 1) in *Tvp23b* ($p = 6.33 = \times 10^{-5}$) on chromosome 11 (Fig. 1a, b). This was predicted to cause exon 1 skipping resulting in a 141 bp deletion, loss of the start codon, and an unpredictable protein product.

To verify that the *Chipotle* allele of *Tvp23b* was causative of the phenotype, we used CRISPR/Cas9 targeting to generate mice bearing a 1 bp frameshift allele of the *Tvp23b* gene. The CRISPR/Cas9 targeted mice (*Tvp23b^{-/-}*) were susceptible to DSS challenge, exhibiting greater than 20% weight loss by day 8 of treatment (Fig. 1c). The *Tvp23b^{-/-}* mice also exhibited increased disease activity manifested by diarrhea, rectal bleeding, as well as colonic shortening (Fig. 1d, e). Colons from the *Tvp23b^{-/-}* mice showed marked histopathological changes characterized by infiltration of lymphocytes and loss of crypt architecture (Fig. 1f). These data confirm that TVP23B is critical and non-redundant for regulating intestinal homeostasis in response to DSS.

To determine whether this sensitivity extended to other forms of colitis, we employed the *Citrobacter rodentium* model of colitis, which mimics human enteropathogenic *Escherichia coli* infection. The oral infection of *Tvp23b^{-/-}* mice with the colonic pathogen *C. rodentium* resulted in severe weight loss and diarrhea (Fig. 1g). Mice failed to clear the bacteria as indicated by an elevated stool burden of the bacteria (Fig. 1h, i). These data demonstrate that TVP23B is necessary for clearing intestinal pathogens.

### Intestinal epithelial intrinsic requirement for TVP23B expression

To understand where the protein is located, we generated a 3XFLAG epitope tagged knock-in mouse using CRISPR/Cas9. In the intestine, *TVP23B* protein is highly expressed in the epithelial cell layer. (Fig. 2a). Immunohistochemical analysis of small intestinal and colonic tissues showed a staining pattern that is limited to the epithelium of these tissues and most prominently in the crypt (Fig. 2b). No staining was observed in the underlying lamina propria, which is rich with immune cells, or smooth muscle containing muscularis layers of the intestines of the epitope tagged mice.

To establish the cell of origin requiring TVP23B in the colitis phenotype, bone marrow chimeric mice were generated. Donors and/or recipients were either CD45.1 or *Tvp23b^{-/-}* (CD45.2) strains. The reconstitution efficiency of donor cells ranged from 94−97% in the peripheral blood, irrespective of genotype. After DSS administration, chimeric CD45.1 recipient mice with *Tvp23b^{-/-}* hematopoietic cells did not exhibit any weight loss, similar to control CD45.1 mice receiving transplants of WT bone marrow (Fig. 2c). Chimeric *Tvp23b^{-/-}* recipient mice, irrespective of donor bone marrow, were not protected from DSS challenge and lost 20% of their initial body weight by day eight. The weight loss coincided with increased rectal bleeding and diarrhea (Fig. 2d) as well as reduced colon length (Fig. 2e). Moreover, the peripheral immune profile of TVP23B-deficient animals was similar in the development of major immune subsets, antibody production, aldehyde production as compared to wildtype littermates (Supplementary Figs. S2 and S3). Taken together, these findings suggested the *Tvp23b^{-/-}* colitis sensitivity phenotype is caused by an epithelial cell intrinsic defect.

To confirm an epithelial intrinsic defect, we generated a conditional allele targeting exon 2 of TVP23B, deletion of which creates a frameshift mutation in the native protein. Mice were crossed to the Villin-Cre background to generate intestinal epithelial specific deletion and tested on DSS. *Tvp23b^{FL/FL}*; Villin-CRE mice were sensitive to DSS displaying enhanced weight loss, increased DAI, and shortened colons compared to *Tvp23b^{FL/FL}* littermate controls (Fig. 2f–h). This data firmly establishes that TVP23B is required in the intestinal epithelium to prevent colitis.

### TVP23B is required for Paneth cell and goblet cell form and function

Given the hematopoietic extrinsic nature of the phenotype in *Tvp23b^{-/-}* animals, we examined the intestinal epithelium in the homeostatic (i.e. non-DSS treated) state. Histologically, an absence of granule containing cells in the ileums of *Tvp23b^{-/-}* mice was observed using PAS staining and immunofluorescent staining for lysozyme (Fig. 3a, b). Ultrastructural analysis revealed smaller dense core granules (DCGs) and a paucity of DCG containing cells in the small intestinal crypt consistent with a loss of Paneth cells (Fig. 3c). We then used mass spectrometry to measure small soluble peptides in the terminal ileum. In *Tvp23b^{-/-}* ileums, 23 alpha-defensins were found to be decreased by at least 2-fold relative to *Tvp23b^{+/-}* levels (Fig. 3d and Supplementary Data 1). This loss of small antimicrobial peptides is consistent with a defect in Paneth cell homeostasis or function. We found that transcripts of Paneth cell specific genes including antimicrobial peptides were markedly reduced in *Tvp23b^{-/-}* ileums, consistent with the observed absence of Paneth cells (Fig. 3e). Collectively, these data demonstrate that TVP23B is essential for the homeostasis of Paneth cells in vivo, and that TVP23B deficiency results in a loss of antimicrobial function by these cells.

Similar examination of the colon showed decreased staining of goblet cell vacuoles with Alcian blue in *Tvp23b^{-/-}* animals, although we detected no significant decrease in absolute number of goblet cells (Fig. 4a). Using bulk RNA-seq of colonic epithelial mRNA, no changes in key goblet cell markers (Muc2, Atoh1, Spdef1, Tff3) were noted, which was consistent with the number of goblet cells being unchanged in the mutant animals (Figure S4). Goblet cells were imaged using electron microscopy and found to have abnormally septate and small mucus containing vacuoles compared to wildtype littermates (Fig. 4b). A similar phenotype was found in goblet cell differentiated organoid cultures with smaller average PAS staining vacuole size (7.9 μm vs 5.3 μm) in the *Tvp23b^{-/-}* cells thus demonstrating that the phenotype is epithelial intrinsic and not dependent on the microbiome (Fig. 4c, d). To elucidate a role in mucin secretion, distal colon was isolated and mucin secretion tested ex vivo. *Tvp23b^{-/-}* colons had a thinner layer of mucus at baseline as well as under homeostatic growth conditions (Fig. 4e, f). LPS stimulation of the colon is known to activate sentinel

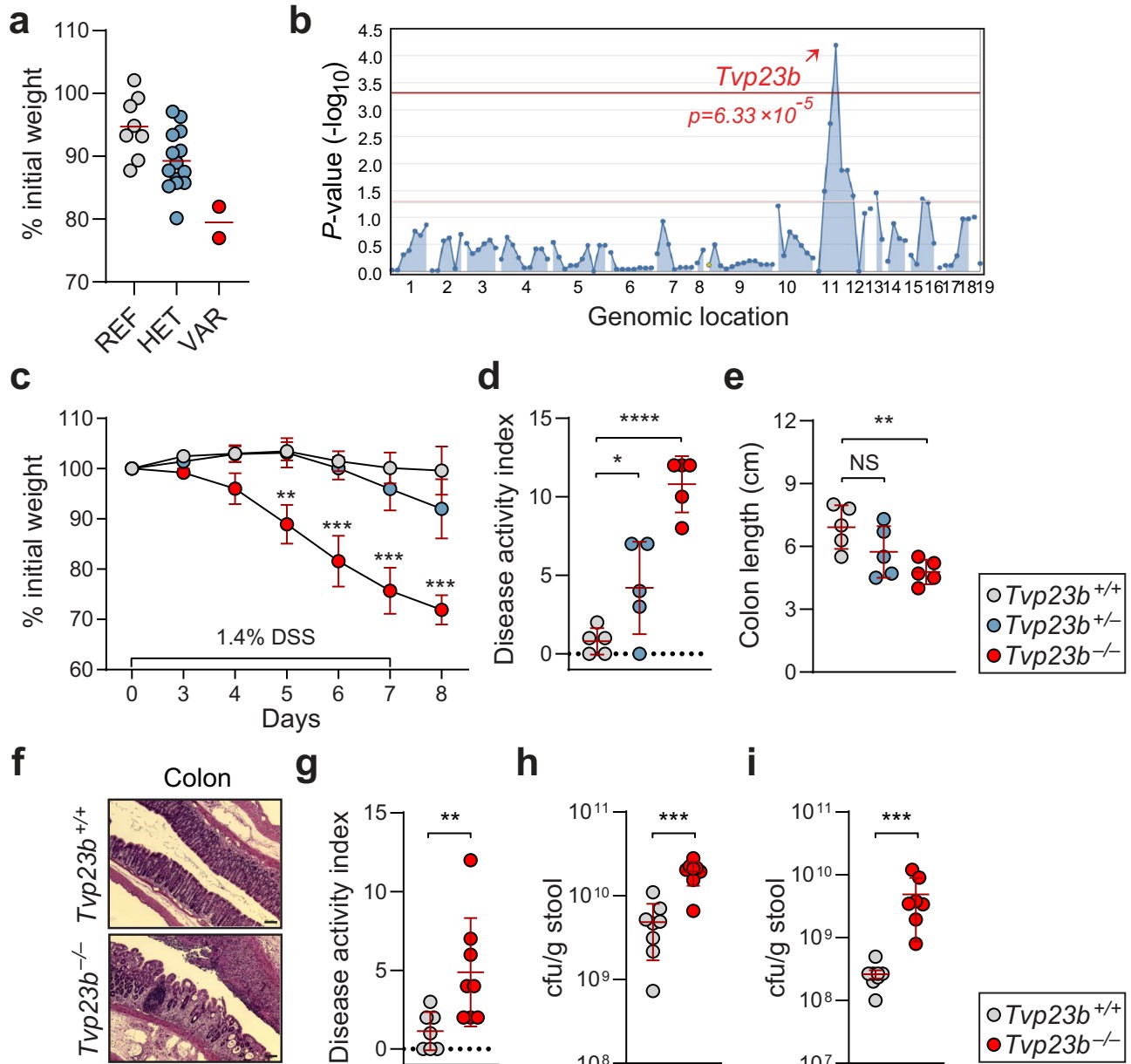

**Fig. 1 | Mapping and validation of Tvp23b in *chipotle* phenotype. a** Percentage of body weight lost on day 7 of DSS treatment plotted per genotype [REF *Tvp23b*+/+ (n = 8); HET *Tvp23b*+/chipotle (n = 13); VAR *Tvp23b*chipotle/chipotle (n = 2)]. **b** Manhattan plot showing P values of association between *chipotle* and mutations identified in *chipotle* pedigree, calculated using an additive model of inheritance. The −log₁₀ P values (y-axis) are plotted versus the chromosomal positions of the mutations (x-axis). Horizontal red and purple lines represent thresholds of P = 0.05 with or without Bonferroni correction, respectively. P values for linkage of mutation in *Tvp23b* with the *chipotle* DSS phenotype are indicated. **c** Weight loss analysis of mice of *Tvp23b*+/+, *Tvp23b*+/−, and *Tvp23b*−/− after 1.4% DSS treatment (n = 5 independent mice for all groups) from CRISPR/Cas9 targeted mice. (**P = 0.0015,

***P < 0.0001) **d, e** Disease activity index and colonic length of individual mice after 7 days of DSS challenge (n = 5 independent mice for each group). (*P = 0.043, **P = 0.015, ****P < 0.0001). **f** Representative Hematoxylin and Eosin staining of *Tvp23b*+/+ and *Tvp23b*−/− colons after 7 days of DSS treatment. Scale bars: 50 μm. **g** Disease activity index of mice 11 days post oral infection with *C. rodentium* (n = 8 independent mice for each group, **P = 0.0064). **h, i** Streptomycin-resistant colony forming units of Feces (***P = 0.0006) and Cecal content (***P = 0.0003) after oral *C. rodentium* infection. Data are expressed as means ± s.d. and significance was determined by two-way ANOVA with Dunnett's multiple comparisons (**c**), one-way ANOVA with Dunnett's multiple comparisons (**d, e**), Mann–Whitney test (**g, h, i**). Data are representative of at least three independent experiments (**c–i**).

goblet cells for further mucin secretion[13]. This response was abrogated in *Tvp23b*−/− colons (Fig. 4g). Overall, these data demonstrate that the function of colonic goblet cells in making a mucous layer is abrogated in *Tvp23b*−/− mice.

**Host microbiome penetrates the mucous layer of TVP23B deficient mice, colonizing the tissue**

To gain a quantitative assessment of putative microbe penetration into the mucus in the colon, fluorescent beads were visualized ex vivo.

Compared to *Tvp23b*+/− colons, *Tvp23b*−/− colons had a smaller distance between the Syto9 stained tissue and the microbe sized beads (Fig. 5a, b). *Tvp23b*−/− colons also had increased bead penetration into the mucous layer compared to heterozygous littermates (Fig. 5c). This was highlighted by increased intestinal permeability to non-absorbable FITC-Dextran (Fig. S5a). These data indicate that the major colonic function of TVP23B may be in regulating the content and formation of mucin containing vacuoles and the secretion thereof. TVP23B deficient animals have more penetrable mucous layers

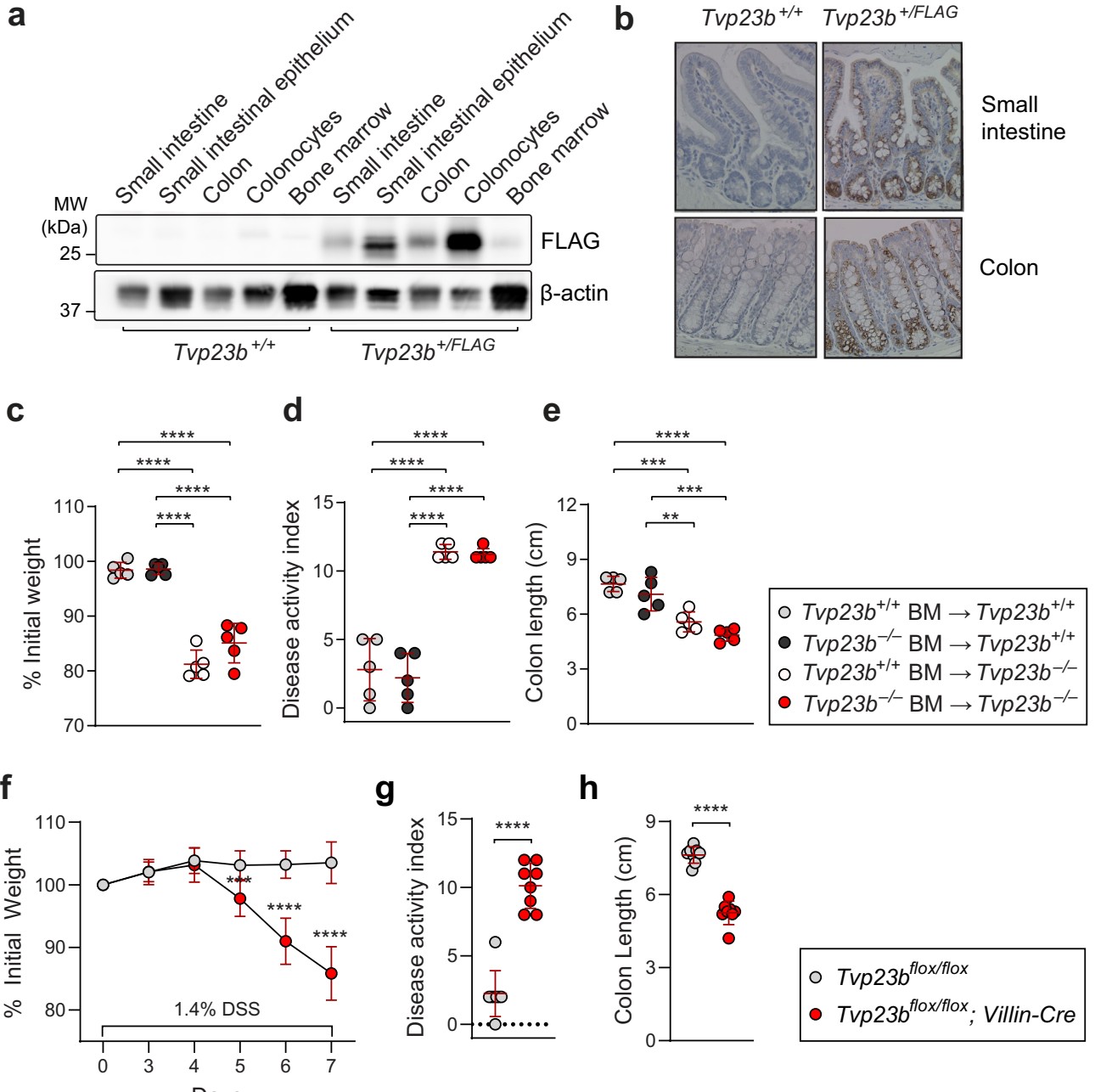

**Fig. 2 | Expression and necessity of TVP23B in the intestinal epithelium.**
**a** Immunoblot analysis of tissues from *Tvp23b+/+* and *Tvp23b+/FLAG* mice.
**b** Representative anti-FLAG immunohistochemical staining of *Tvp23b+/+* and
*Tvp23bFLAG/FLAG* distal small intestine and distal colon tissue. Bone marrow chimeras
were generated, and percentage of initial body weight (**c**), disease activity index (**d**)
and colon length (**e**) were determined after 1.4% DSS administration (*n* = 5 inde-
pendent for all groups; **P = 0.0051, ***P = 0.003, ****P < 0.0001). **f**–**h** Intestinal

specific deletion of *Tvp23b* using floxed allele (*Tvp23bFL/FL*) on the Villin-Cre back-
ground were tested for DSS sensitivity, assessing weight loss, disease activity and
colon length (*n* = 8 individual mice for each genotype, ***P = 0.0005,
****P < 0.0001). Data are expressed as means ± s.d. and significance was determined
by one-way ANOVA with Dunnett's multiple comparisons (**c**–**e**), two-way ANOVA (**f**)
and unpaired Student *t*-test (**g**, **h**). Data are representative of at least 3 independent
experiments.

compared to littermate controls, and coupled with the loss of
Paneth cell antimicrobial peptides, highlights a possible excessive
host−microbe interaction.

Using 16S FISH, the segregation of epithelial cells from microbial
flora was visualized, demonstrating the sterile zone between host and
microbe was greatly decreased in *Tvp23b−/−* colons (Fig. 5d, e). Co-
staining of bacteria and the mucus plumes indicated a large number of
bacteria residing within the thinner mucus layer of *Tvp23b−/−* colons
(Fig. 5f). Quantitative 16S PCR revealed an increase in the number of
colon tissue-associated bacteria in TVP23B deficient animals,

indicating increased penetration and colonization into the tissues
(Fig. 5f). 16S sequencing of mucosal associated bacteria were similar
but revealed a preponderance of *Helicobacter* species within the tissue
of *Tvp23b−/−* mice (Fig. S5b). No differences, however, were noted
between *Tvp23b−/−* and *Tvp23b+/−* mice in the quantity of bacteria in
fecal pellets (Fig. S5c). The control and levels of segmented fila-
mentous bacteria (SFB) are a key contributor to intestinal inflamma-
tory activation[14]. In *Tvp23b−/−*, however, no significant increase in SFB
was observed in tissue-associated bacteria (Fig. S5d). Overall, these
data indicate that TVP23B is critical for maintaining host−microbial

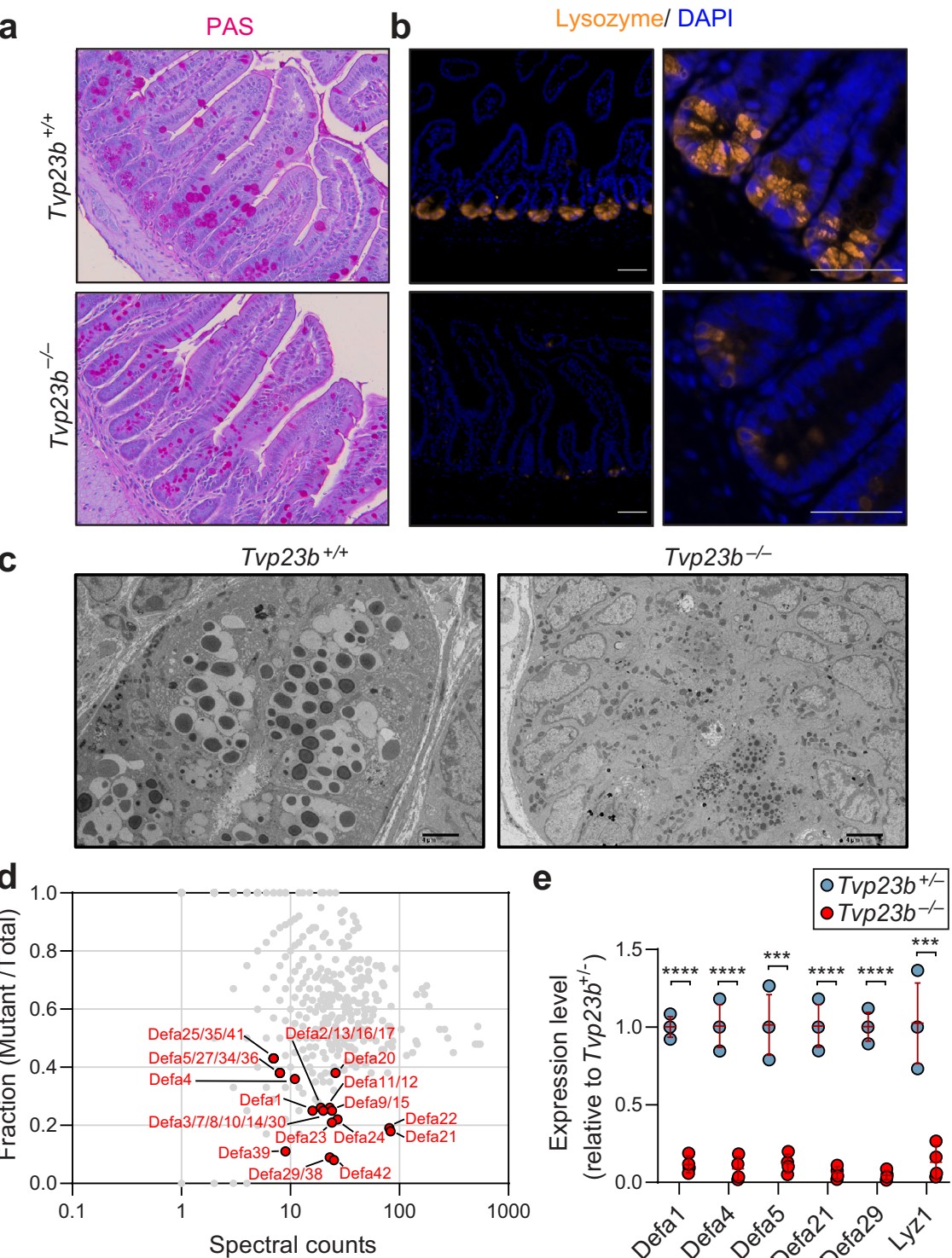

**Fig. 3 | Paneth cell and antimicrobial peptide deficiency in TVP23B-deficient small intestine. a** Representative periodic acid Schiff (PAS) staining and electron microscopy from *Tvp23b*⁺/⁺ and *Tvp23b*⁻/⁻ terminal ileums. **b** Lysozyme immunofluorescent staining of *Tvp23b*⁺/⁺ and *Tvp23b*⁻/⁻ tissue. **c** Representative electron microscopy from *Tvp23b*⁺/⁺ and Tvp23b⁻/⁻ terminal ileums. **d** Composite plot of relative abundance as determined by mass spectrometry from small soluble peptide fraction of *Tvp23b*⁺/⁺ and *Tvp23b*⁻/⁻ terminal ileums (*n* = 3 for each genotype). Normalized protein level [mutant sample/(mutant sample) + (HET sample)] is plotted on the *Y*-axis. Points at *Y* = 1 denote proteins exclusively identified in the mutant sample; points at *Y* = 0 denote proteins exclusively identified in the HET sample. Protein abundance correlates with spectral count **e** Relative mRNA expression of Paneth cell specific genes, *alpha-defensin-1, alpha-defensin-4, alpha-defensin-5, alpha-defensin-21, alpha-defensin-29, lysozyme1* (*n* = 4 ileal samples from independent mice; ***P < 0.001, ****P < 0.0001). Data are expressed as means ± s.d. and significance was determined by unpaired Student *t*-test (**e**). Data is representative of at least 3 independent experiments.

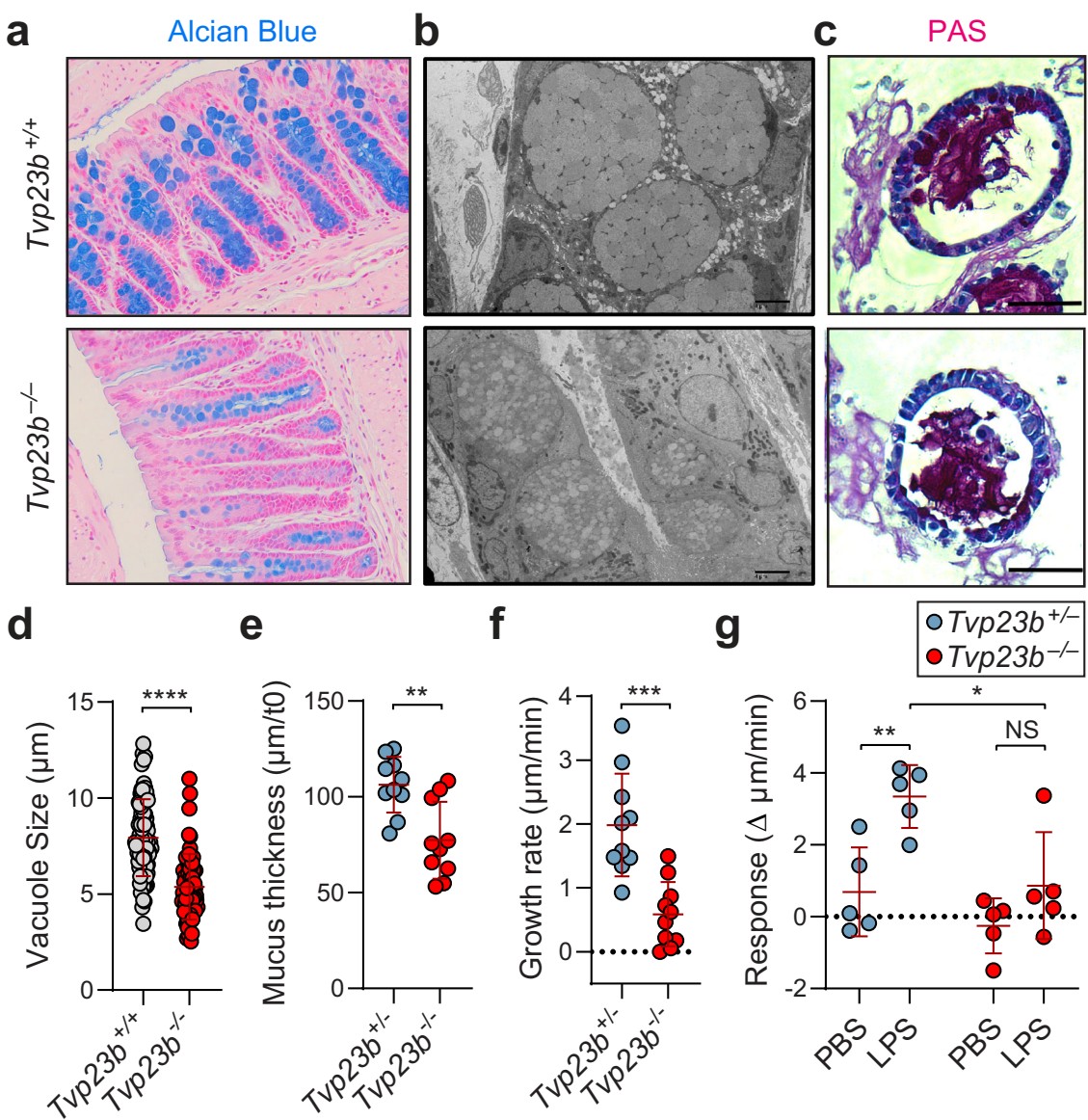

**Fig. 4 | Abnormal goblet cell morphology and mucus production.**
**a**, **b** Representative Alcian-Blue staining and electron microscopy of distal colon from *Tvp23b*⁺ᐟ⁺ and *Tvp23b*⁻ᐟ⁻ animals. **c**, **d** Representative PAS stained colonoids (20×) from *Tvp23b*⁺ᐟ⁺ and *Tvp23b*⁻ᐟ⁻ animals. Quantitation of PAS+ vacuole size from colonoids ($n = 80$ and $n = 73$ individual Goblet cells derived from 3 independent *Tvp23b*⁺ᐟ⁺ mice and 3 *Tvp23b*⁻ᐟ⁻ mice, ****$P < 0.0001$). **e**, **f** Ex vivo mucus thickness and growth rate from *Tvp23b*⁺ᐟ⁻ and *Tvp23b*⁻ᐟ⁻ colons ($n = 10$ independent mice per group **$P = 0.0017$, ***$P = 0.0002$). **g** Ex vivo induction of mucus by lipopolysaccharide in *Tvp23b*⁺ᐟ⁻ and *Tvp23b*⁻ᐟ⁻ colons ($n = 5$ independent mice per group, *$P = 0.0149$, **$P = 0.0091$). Data are expressed as means ± s.d. and significance was determined by unpaired student *t*-test (**d**) one-way ANOVA with Dunnett's multiple comparisons (**e**, **f**) and two-way ANOVA (**g**).

segregation. Deficiency of TVP23B leads to increased proximity and penetration by commensal and pathogenic microbes to the host epithelium.

## TVP23B regulates epithelial cell protein glycosylation

We observed a thinner mucus layer with decreased lectin staining in *Tvp23b*⁻ᐟ⁻ colons relative to those in WT colons (Fig. 6a). This was despite the normal mucus proteome and normal number of MUC2 molecules per volume of mucus in *Tvp23b*⁻ᐟ⁻ colons (Fig. 6b, c). Immunoblots of colonocytes using an anti-Muc2 antibody, however, revealed a decreased molecular weight *Tvp23b*⁻ᐟ⁻ samples, indicating potentially decreased glycosylation or other posttranslational modification (Fig. 6d). To directly examine glycosylation, we undertook unbiased glycomic analysis of TVP23B deficient colonocytes (Supplementary Data 2). A ninefold decrease in core-3 O-glycosylation was observed in *Tvp23b*⁻ᐟ⁻ colonocytes (Fig. 6e). O-glycosylation cores 1–4

are determined by the initial sugar moiety added to the protein (Fig. S6a). Smaller decreases in sialylation of O-glycans were also present (Fig. S6b). Sialylation of intestinal epithelial cell proteins including mucin has been shown to have an immunomodulatory affect[15–17]. Core-3 O-glycosylation is the major O-glycan form on intestinal mucins and is needed for host–microbe separation[18]. Thus, the decrease in MUC2 molecular weight and altered glycome in *Tvp23b*⁻ᐟ⁻ colonocytes suggest that TVP23B is required for intestinal mucus formation through its support of glycosylation.

## TVP23B forms a complex with the Golgi protein YIPF6

Similar to yeast TVP23, mammalian TVP23B localized strongly with the Golgi in cellular fractionation experiments (Fig. 7a). Yeast TVP23 was also noted to have several direct binding partners, including the homolog of mammalian YIPF6[19]. Mammalian YIPF6 had previously been implicated in a DSS sensitivity screen as a regulator of intestinal

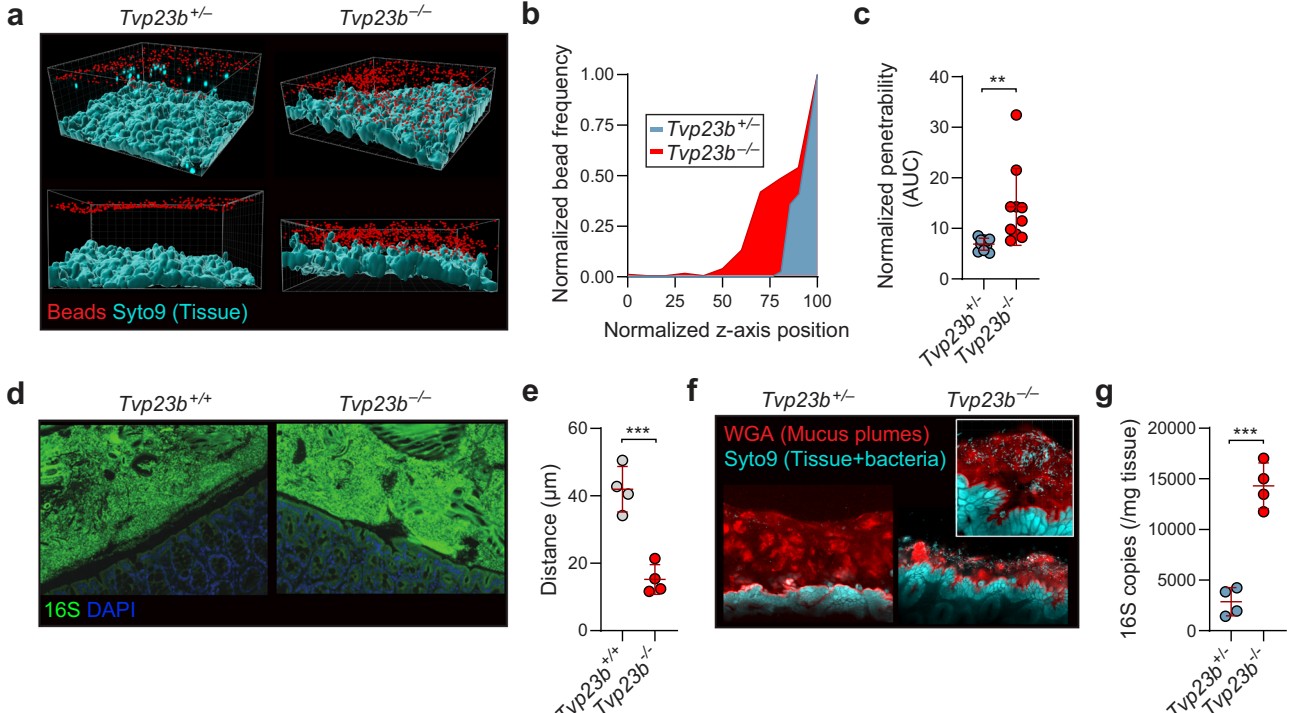

**Fig. 5 | Defective host–microbe segregation. a** Representative confocal z-stacks showing x/z-axis cross sections of tissue obtained from *Tvp23b+/–* and *Tvp23b–/–* littermates; tissue (blue), 1-μm beads (red). **b, c** Normalized distances and penetrability to beads from colonic tissue (*n* = 8 per genotype, **P = 0.0072). **d, e** Visualization and quantification of microbiota localization relative to colonic mucosal surface using fluorescent in situ hybridization. **d** Sections were hybridized to a probe that recognizes 16S rRNA of all bacteria (green) and counterstained with DAPI (blue). **e** Average distance of inner mucus layer quantified from 16S FISH to close epithelium in at least 20 sections per mouse (*n* = 4 per genotype, ***P = 0.0006). **f** Representative mucus staining with lectin wheat germ agglutinin (WGA) and counterstains for tissue and bacteria (Syto9, blue). **g** Mucosa-associated bacteria were quantified by qPCR of 16S rRNA gene copy number in distal colon (*n* = 4 per genotype, ***P = 0.0001). Data (**c, e, g**) are expressed as means ± s.d. and significance was determined by unpaired Student *t*-test.

homeostasis and deficiency results in a similar cellular phenotype in Paneth and goblet cells as observed in the TVP23B-deficient animals[20]. We initially sought to confirm this interaction by co-immunoprecipitation. We immunoprecipitated HA-tagged TVP23B with FLAG-tagged YIPF6 from transiently co-transfected cells (Fig. S7a). Additionally, we detected endogenous TVP23B bound to YIPF6 in goblet cells lines (HT29-MTX and LS174T) that were transduced with FLAG-tagged YIPF6 (Fig. 7b). Interestingly, *Yipf6*-deficient animals (*Yipf6Klz/Y*) displayed significantly decreased levels of TVP23B in colonocytes, indicating that YIPF6 is required for TVP23B expression (Fig. 6c). This stabilization of TVP23B was also present in 293T cells upon YIPF6 Co-transfection (Fig. S7b). These data indicate that TVP23B and YIPF6 bind each other in vivo and that YIPF6 is required for TVP23B stability in the cell.

### TVP23B and YIPF6 regulate the Golgi Proteome

TVP23B and YIPF6 have both been implicated in intracellular transport to the Golgi[21,22]. To test whether lack of these proteins critically changes the composition of the Golgi, an unbiased mass spectrometric approach was taken to examine the Golgi proteome. Golgi complexes were isolated from the colonic epithelia of *Tvp23b+/+* and *Tvp23b–/–* animals using sucrose sedimentation and analyzed by mass spectrometry (Fig. 7d). A total of 117 proteins were decreased at least 50%, and 308 proteins were increased twofold or more in *Tvp23b–/–* Golgi compared to WT Golgi (Supplementary Data 3). A similar approach was taken for analysis of *Yipf6+/Y* and *Yipf6Klz/Y* animals, yielding 179 proteins decreased and 196 proteins increased in *Yipf6Klz/Y* Golgi compared to control Golgi (Fig. 7e and Supplementary Data 3). Among the proteins with reduced expression in either mutant, 25 proteins were decreased in both *Yipf6Klz/Y* and *Tvp23b–/–* Golgi, of which 10 have described

functions as glycosyltransferases (Fig. 7f). Other reduced proteins include proteins critical for protein reduction (QSOX1) as well as ion channels (SLC35A1, ATP2C2) are also known to be critical for glycosylation[23,24]. B3gnt6 is required for core-3 glycosylation of intestinal mucins[18] and was lost completely in both *Yipf6Klz/Y* and *Tvp23b–/–* Golgi (Fig. 7d, e). The Golgi fraction displayed a higher molecular weight, presumably glycosylated, B3GNT6, which was lost in mutant animals (Fig. 7g, h). This protein loss is in contrast to another Golgi localized glycosyltransferase, GALNT4, which is unchanged in the mutant (Fig. S8). Taken together, these data suggest that TVP23B and YIPF6 cooperate in regulating the composition of key Golgi enzymes, including glycosylation enzymes needed for intestinal barrier function.

## Discussion

Using ENU based germline forward genetics and CRISPR/Cas9 germline targeting, we demonstrated that a damaging mutation in the Golgi resident protein TVP23B renders mice susceptible to DSS-induced and infectious colitis, which is presumably due to the same mucus defect. *Tvp23b–/–* colons have a thinner, more penetrable mucus layer, which impacts host–microbe interactions. This appears to result from a primary defect in glycosylation, and specifically core-3 linked O-glycosylation and sialylation. Molecularly, TVP23B binds and is stabilized by YIPF6, another Golgi protein required for Paneth cell and goblet cell homeostasis. These proteins work in concert to regulate the Golgi proteome, including B3GNT6, which is necessary for core-3 O-glycosylation of the dominant gel forming mucin, MUC2[18]. Overall, these data demonstrate the necessity of TVP23B for maintenance of intestinal homeostasis and host–microbe interactions.

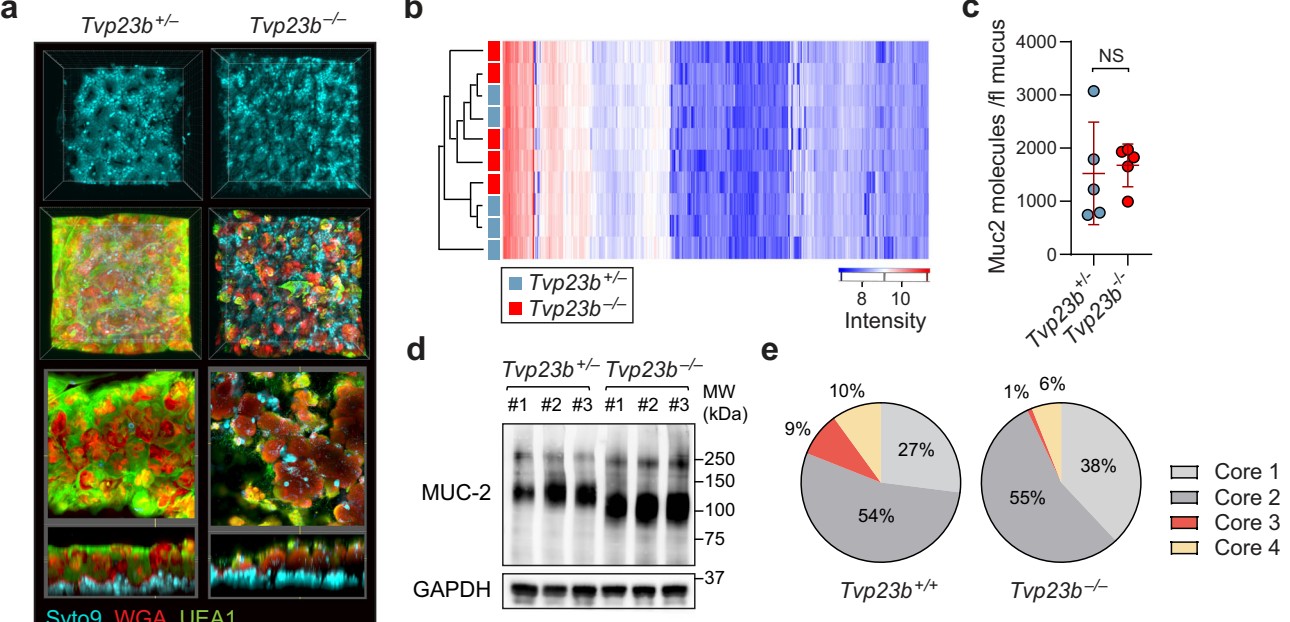

**Fig. 6 | Glycosylation defect of *Tvp23b⁻/⁻* colonocytes. a** Representative staining of mucous layer from *Tvp23b⁺/⁻* and *Tvp23b⁻/⁻* colons. **b** Proteomic analysis of colonic mucus layer composition. **c** Muc2 concentration as determined per femtoliter of mucus. (*n* = 5 independent mice per group) **d** Immunoblot of Mucin-2 in colonocytes from 3 separate *Tvp23b⁺/⁻* and *Tvp23b⁻/⁻* animals. **e** Representative core linkage analysis of O-glycans in *Tvp23b⁺/⁺* and *Tvp23b⁻/⁻* colonocytes from three separate experiments with two mice per genotype. Data (**c**) are expressed as means ± s.d. and significance was determined by unpaired Student *t*-test.

Remarkably, both our Golgi proteomics as well as our glycomic analysis converge on the glycosyltransferase B3GNT6. A total of 10 glycosyltransferases were found to be decreased in common between TVP23B and YIPF6-deficient Golgi, speaking to the broad affect TVP23B has in regulating intestinal glycosylation. Other proteins decreased in TVP23B deficient Golgi including QSOX1 and Slc35a1 are known also to affect glycosylation, although not acting as direct transferases[23,24]. The overall effect of TVP23B deficiency is analogous to congenital disorders of glycosylation (CGD). CGDs will often have broader effects including neurologic, growth, and hematopoietic defects, but also have been found to include profound enteropathies[25]. A recent report has linked a monogenic form of human inflammatory bowel disease to glycosylation defects in the colon[15]. Our study adds to the glycosylation canon by increasing our understanding of the glycosyltransferase distribution in the mammalian intestine.

An open question is understanding the loss of Paneth cells in the *Tvp23b⁻/⁻* and *Yipf6^{Klz/Y}* animals. Mucin-2 surrounds the dense core granules found in Paneth cells and Muc2-deficiency leads to reduced Paneth cell number[26]. One could postulate that the loss of proper glycosylation leads to increased unfolded proteins, resulting in cell death as animals deficient in this pathway have been shown to lack Paneth cells[27,28]. In the future, purification of Paneth cells from these animals or Paneth cell specific ablation may be illuminating for this mechanism.

TVP23B mice do not display a general defect in secretion in all cell types and the phenotype thus far is restricted to intestinal secretory cells. Expression of TVP23B is found in many immune cell subsets, although no overt phenotype was found. This may be due to a number of factors including redundancy and expression patterns of TVP23B. Mice have both TVP23A and B while humans have TVP23A-C. These multiple homologs may serve redundant functions or compensatory roles in different cell types and is an area worthy of investigation.

The identification of YIPF6 as a binding partner for TVP23B is particularly notable. Both proteins were identified via ENU forward genetic screens, and mutations resulted in the same phenotype of DSS sensitivity as well as a loss of Paneth cell and goblet cell

morphology[29,30]. This molecular interaction appears to be conserved down to yeast as yeast TVP23 binds yeast YIPF4 (homolog of mammalian YIPF6). The molecular function of this complex appears to be in cargo transport to the Golgi resulting in a change in its protein composition. Given our Golgi proteomic findings with partially non-overlapping protein sets, it is possible that YIPF6 has both TVP23B dependent and independent functions.

Both YIPF6 and TVP23B have broad effects on the protein composition of the Golgi, but a common striking loss of many glycosyltransferases (DAVID analysis *p* value $3.3 \times 10^{-13}$). This could reflect many possible mechanisms including potential roles as cargo adapters, binding, packing and transporting specific cargoes into vesicles. Previous reports have linked YIPF6 to specific cargoes as well as Golgi reformation and these are plausible mechanisms of TVP23B action[22]. Alternatively, these proteins could be important for the localization and function of key ion transport channels. Golgi integrity as well as mucin packing are known to be dependent on the concentration of divalent cations (i.e. $Mg^{2+}$ and $Ca^{2+}$) as well as on the pH of the compartment[31,32]. The TVP23B mutant proteome had reductions in several solute channels (e.g. ATP2C2), which make this an attractive hypothesis. The determination of pH and $Ca^{2+}$ concentrations of TVP23B deficient Golgi will be of interest and an area of future investigation.

Inflammatory bowel disease has long been associated with defects in barrier function of particular mucins, which are dependent on glycosylation for their gel forming properties[33–35]. Multiple mutations in human *TVP23B* have been found. In the Ashkenazi Jewish population a heterozygous intronic mutation is associated with ulcerative colitis, a disease known to be exacerbated by mucin defects[36]. As noted in our ENU screening and CRISPR/Cas9 knockout mice, heterozygous *Tvp23b* mutant animals do have a minor increase in colitis sensitivity. Given its non-coding location, it is unclear if this human mutation affects expression of *TVP23B* or splicing of the primary transcript. Although appealing, caution should be exercised in drawing a conclusion from a heterozygous intronic mutation on a heterogenous background. Further studies in individuals bearing strongly deleterious mutations

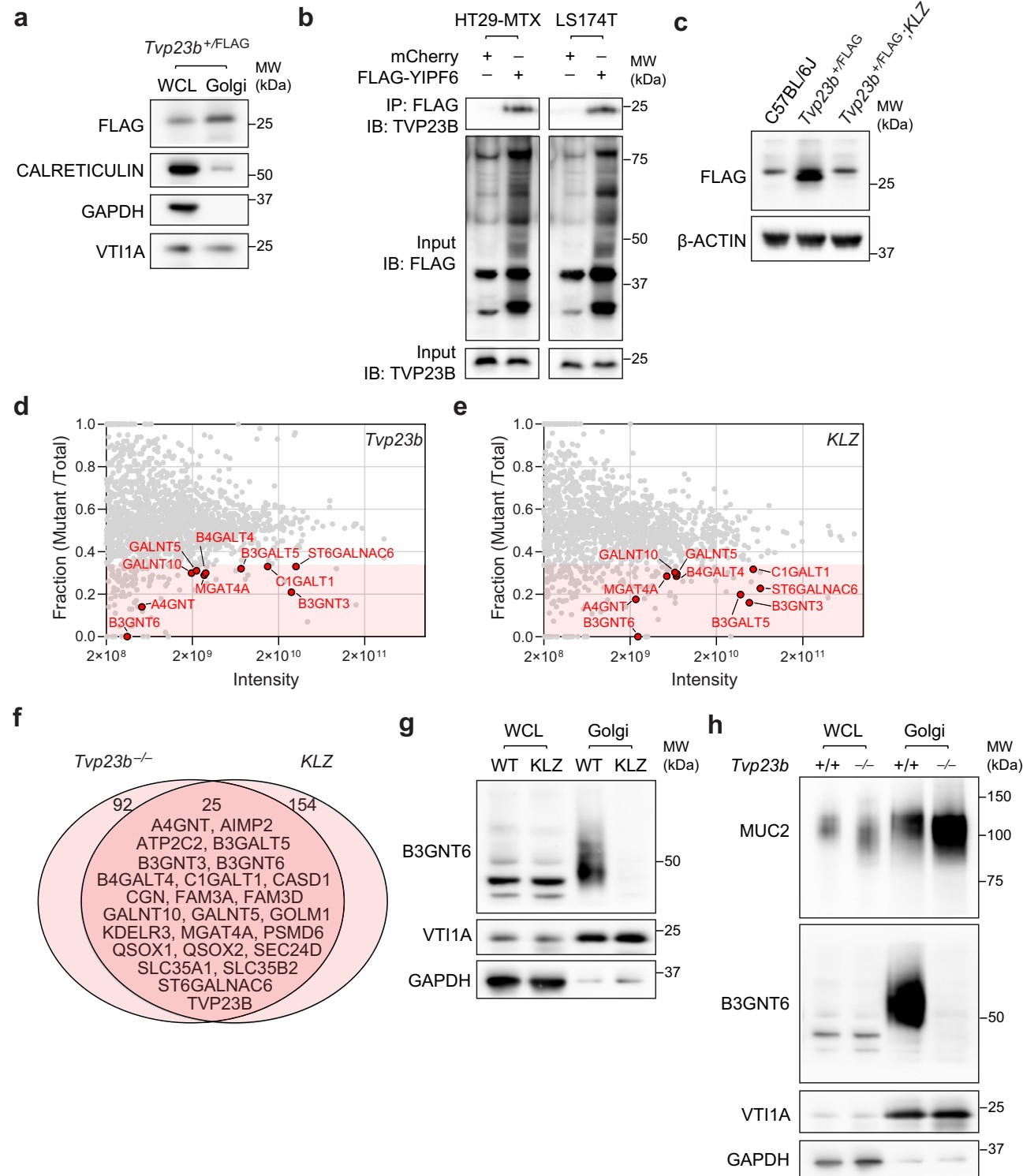

**Fig. 7 | Golgi Proteomics of TVP23B and YIPF6-deficient colonocytes.**
**a** Immunoblot showing Golgi enrichment of FLAG-TVP23B in *Tvp23b*$^{+/FLAG}$ mice.
**b** Co-immunoprecipitation of YIPF6 and TVP23B from transiently transfected
HT29-MTX and LS174T human goblet cell lines. **c** Destabilization of FLAG-TVP23B
protein in YIPF6-deficient hemizygous male mice (*Yipf6*$^{Klz/Y}$). **d** Composite pro-
teome of Golgi enriched fractions from *Tvp23b*$^{-/-}$ colonocytes compared to litter-
mate control. **e** Composite proteome of Golgi enriched fractions from *Yipf6*$^{klz/y}$
colonocytes compared to littermate control. In **d** and **e**, proteins identified by LC-
MS/MS. Normalized protein level [mutant sample/(mutant sample)+(WT sample)]

is plotted on the *Y*-axis. Points at *Y* = 1 denote proteins exclusively identified in the
mutant sample; points at *Y* = 0 denote proteins exclusively identified in the WT
sample. Protein abundance correlates to intensity. Red shading represents less than
0.33 fraction of mutant/total. **f** Venn diagram of proteins with twofold reduction in
intensity in both *Tvp23b*$^{-/-}$ and *Yipf6*$^{klz/y}$ samples. **g**, **h** Immunoblot analysis of
B3GNT6 in Golgi enriched fractions from *Tvp23b*$^{-/-}$ and *Yipf6*$^{klz/y}$ samples. All Golgi
proteomics were performed three times with at least four mice per genotype in
each sample. All immunoblots are representative of at least three independent
experiments.

(nonsense and splice) would be needed to truly establish a causal role in human disease.

In summation, using forward genetics, we have identified an essential and non-redundant role for *Tvp23b* in host–microbe segregation and intestinal homeostasis maintenance in vivo by controlling intestinal glycosylation.

## Methods

### Ethics statement

All experimental procedures using mice were approved by the Institutional Animal Care and Use Committee of the University of Texas Southwestern Medical Center and were conducted in accordance with institutionally approved protocols and guidelines for animal care and use. All mice were bred and housed at the University of Texas Southwestern Medical Center animal facility in accordance with institutionally approved protocols.

### Mice

Eight to ten week old C57BL/6J were purchased from The Jackson Laboratories. ENU mutagenesis was performed[6]. For the DSS-induced colitis induction, mice received 1.4% (wt/vol) DSS in the drinking water for seven days followed by three days off DSS. Body weight was recorded daily and reported as the amount of weight loss from the pre-treatment weight. For the *C. rodentium* infection, animals were pretreated with antibiotics (ampicillin, vancomycin, neomycin and metronidazole) for a minimum of 3 days. They were then given $10^9$ CFU of streptomycin-resistant *C. rodentium* and weighed daily. Stool was collected and serial dilutions were performed for enumeration of *C. rodentium* CFUs. Disease activity index score is a composite score of weight loss, stool bleeding and stool consistency determined as previously described. Briefly: weight loss: 0, No loss. 1, 1–10% loss of body weight. 2, 10–15% loss of body weight. 3, 15–20% loss of body weight. 4, >20% loss of body weight; stool consistency: 0 (normal), 2 (loose stool), and 4 (diarrhea); and bleeding: 0 (no blood), 1 (hemoccult positive), 2 (hemoccult positive and visual pellet bleeding), and 4 (gross bleeding and/or blood around anus). All mice were housed in University of Texas Southwestern vivarium. All mouse experiments were performed using both male and female mice. Mice were housed at 22 °C with a 12-h light/12-h dark cycle. Animals were fed ad libitum with standard chow diet (2016 Teklad Global 16% Protein Rodent Diet) and fresh autoclaved water.

### Generation of the Tvp23b$^{-/-}$ and 3XFLAG-tagged mouse strain using the CRISPR/Cas9 system

To generate the *Tvp23b$^{-/-}$ and 3XFLAG* mouse strain, female C57BL/6J mice were superovulated by injection of 6.5 U pregnant mare serum gonadotropin (PMSG; Millipore), followed by injection of 6.5 U human chorionic gonadotropin (hCG; Sigma-Aldrich) 48 h later. The superovulated mice were subsequently mated overnight with C57BL/6J male mice. The following day, fertilized eggs were collected from the oviducts and in vitro–transcribed Cas9 mRNA (50 ng/μl) and *Tvp23b* small base-pairing guide RNA (KO: 50 ng/μl; 5′-CTCACTGTTTGATGCAGAAG-3′) or (3XFLAG: 50 ng/μl; 5′-CGTCATGTTGTCGCAGGTGA-3′) injected into the cytoplasm or pronucleus of the embryos. The conditional knockout allele (CKO) was generated using two gRNA (5′-GTAC-TAGGGATTCACAGCGT-3′ and 5′-TCATATCAAAAAGTGGGACA-3′) and a single DNA template encoding exon 2 of TVP23B with flanking LoxP sites. The injected embryos were cultured in M16 medium (Sigma-Aldrich) at 37 °C in 5% CO$_2$. For the production of mutant mice, two-cell stage embryos were transferred into the ampulla of the oviduct (10–20 embryos per oviduct) of pseudo-pregnant Hsd:ICR (CD-1) female mice (Harlan Laboratories).

### Generation of bone marrow chimeric mice

Recipient mice were lethally irradiated with two 7-Gy exposures to X-irradiation administered 5 h apart[11]. Femurs derived from donor *Tvp23b$^{+/+}$* or *Tvp23b$^{-/-}$* mice were flushed with phosphate-buffered saline (PBS) using a 25 G needle. The cells were centrifuged at $700 \times g$ for 5 min, and cells were resuspended in 1 mL PBS and transferred into 1.5-mL Eppendorf tubes and kept on ice. Bone marrow cells from *Tvp23b$^{+/+}$* or *Tvp23b$^{-/-}$* mice was transferred into the indicated recipient mice through intravenous injection. For 4 weeks after injection, mice were maintained on antibiotics. Analysis of DSS colitis susceptibility was performed 10 weeks after irradiation and reconstitution.

### Cell culture, transfection, and infection

The Lenti-X 293T cells (Takara #632180) were grown at 37 °C in DMEM (Sigma)/10% (v/v) FBS (Gibco)/1% antibiotics (Life Technologies) in 5% CO2. Transfection of plasmids was carried out using Lipofectamine 2000 (Life Technologies) according to the manufacturer's instructions. Cells were harvested between 36 and 48 h post transfection. The HT29-MTX-E12 cells (Sigma-Aldrich 12940401) and LS174T cells (ATCC CL187) were grown at 37 °C in DMEM (Life Technologies)/10% (v/v) FBS (Gibco)/1% antibiotics (Life Technologies) in 5% CO2. Infections of HT29-MTX-E12 cells and LS174 cells were carried out using a 3rd generation lentiviral system packaged in Lenti-X 293 T cells.

### Crypt isolation

Colonic crypts were isolated as previously described[5]. Briefly, colons were isolated from mice and stool removed from the lumen. Colons were then cut to 5–10 mm pieces and incubated at room temperature for 30 min in PBS containing 5 mM EDTA.

### Intestinal organoid culture

Organoids were cultured from colonic crypts using IntestiCult™ Organoid Growth Medium (Mouse) according to the manufacturer's instructions. Colon organoids were differentiated toward goblet cells by addition of γ-secretase inhibitor DAPT (10 μM) and Wnt inhibitor IWP-2 (10 μM) for 7 days. Colonoids were embedded in low melt agarose prior to sectioning.

### Immunoprecipitation and western blot

Cells were lysed with NP40 lysis buffer with glycerol (50 mM Tris–Cl (pH 8.0), 150 mM NaCl, 1% (v/v) Nonidet P-40, 5% (v/v) glycerol, and protease inhibitors). Immunoprecipitation was performed using anti-FLAG M2 magnetic beads (Sigma) for 1 h at 4 °C, and the proteins were eluted with 150 μg/mL 3× Flag at 4 °C for 1.5 h. Protein concentrations were measured using a BCA assay (Pierce). Samples were loaded onto NuPAGE 4–12% Bis-Tris protein gels (Thermo Fisher Scientific), transferred to nitrocellulose membranes (Bio-Rad), blotted with the primary antibody at 4 °C overnight and the secondary antibody for 1 h at room temperature, and then visualized by chemiluminescent substrate (Thermo Fisher Scientific).

### Electron microscopy

Mice were exsanguinated with 0.9% saline followed by perfusion fixation with 4% paraformaldehyde, 1.5% glutaraldehyde, and 0.02% picric acid in 0.1 M cacodylate buffer and colons and small intestines were dissected and cut into concentric circles. Tissues were incubated overnight at 4 °C and processed using a standard protocol by the UT Southwestern EM core. Tissues mounted on grids were imaged using a JOEL 1400+ microscope.

### Histology and immunostaining

Freshly isolated colons and small intestines were Swiss-rolled, fixed in formalin, and embedded in paraffin. H&E, PAS, and AB staining were conducted using a standard protocol by the UT Southwestern Histology core. For immunostaining, sections were deparaffinized and rehydrated through an ethanol gradient. Antigen retrieval was performed by boiling of slides for 15 min in a citrate-buffered solution (DAKO). Sections were washed with PBS 3X for 5 min, and then blocked

in 10% bovine serum albumin (BSA) in PBS. Primary antibodies were diluted in 5% BSA in PBS. Sections were incubated overnight at 4 °C and then washed 3× for 10 min. Slides were then incubated for 1 h in Alexa fluor antibodies (1:1000) in 5% BSA in PBS and washed 3× for 10 min. Slides were mounted in Prolong Antifade Gold and visualized using a Zeiss LSM880 confocal microscope. All measurements were made using ImageJ software.

## Glycan analysis

Colonocytes were isolated from *Tvp23b*[+/+] and *Tvp23b*[−/−] mice by EDTA chelation. Each sample represents pooled colonocytes from 2 mice and 3 samples per genotype were analyzed. Samples were homogenized in 50 mM ammonium bicarbonate. Samples were denatured using New England Biolabs (NEB) Denaturing buffer and incubated at 45 °C for 30 min. They were then gradually desalted using 10 kDa cutoff spin filters by centrifuging at 14,000 × *g* for 15 min and washed using 400 μL of 50 mM ammonium bicarbonate. The sample remaining in the filter was transferred to a clean Eppendorf tube and homogenized in 500 μL 50 mM ammonium bicarbonate buffer by probe sonicating. The N-glycans were released from proteins by adding PNGase F at 37 °C for 24 h. The released N-glycans were filtered off using 10 kDa cutoff spin filter, purified using C18 cartridge, and were lyophilized. O-glycoproteins from top of the spin filter proceeded to β-elimination. O-glycoproteins that did not flow through 10 kDa spin filter were lyophilized and then dissolved in 50 mM NaOH. 19 mg/250uL NaBH₄ in 50 mM NaOH was added, and the solution was heated at 45 °C for 18 h. Samples were then cooled to room temperature and neutralized by adding 10% acetic acid dropwise. Samples were then passed through a DOWEX H⁺ resin column and a C18 cartridge. Samples were then lyophilized, after which borates were removed using a solution of 9:1 methanol to acetic acid under a stream of N₂. Released N & O-glycans were permethylated by using methyl iodide and DMSO/NaOH mixture. The dried glycans were re-dissolved in methanol and profiled by MALDI-TOF and LC-MS/MS. Dried, permethylated glycans were re-dissolved in a solution of 100 μL of water and 100 μL of methanol for a total volume of 200 μL. Samples were then run on a Themo Fisher Orbitrap Fusion Tribrid tandem MS coupled to an Ultimate 3000 RSLCano ystem. Eight microliters of both N- and O-glycans were injected into LC−MS and ran in the low to high organic solvent gradient for 72 min. 0.5 μg of Xylopentose was used for O-glycans quantification. 0.5 μg of Xylohexose was used for N-glycans quantification.

## Mass spectrometry-based profiling of the mucus proteome

Mucus proteomics were performed as previously described[37]. Briefly, samples were collected ex vivo from intestinal tissues mounted in horizontal perfusion chambers. Mucus was aspirated from the mucosal surface using Maximum Recovery pipette tips (Axygen), mixed with 2x complete protease inhibitor cocktail (Merck). Mucus was reduced overnight in 6 M guanidinum hydrochloride, 0.1 M Tris/HCl (pH 8.5), 5 mM EDTA, 0.1 M DTT (Merck) followed by filter aided sample preparation adapted from a previously developed protocol using 10 kDa cutoff filters (Pall Life Sciences). Proteins were alkylated with iodoacetamide (Merck) and sequentially digested with LysC (Wako) and trypsin (Promega) on the filter. Peptides were cleaned with StageTip C18 columns prior to MS analysis. NanoLC−MS/MS was performed on an EASY-nLC 1000 system (ThermoFisher), connected to a QExactive Hybrid Quadrupole-Orbitrap Mass Spectrometer (ThermoFisher) via a nanoelectrospray ion source. Peptides were separated using an in-house packed reverse-phase C18 column with a 60-min 4–32% acetonitrile gradient. Mass spectra were acquired from 320–1600 *m/z* at resolution 70,000, and the 12 peaks with highest intensity were fragmented to acquire the tandem mass spectrum with a resolution of 35,000 and using automatic dynamic exclusion. Proteins were identified using MaxQuant (v1.5.7.4).

## Enrichment of Golgi

Golgi isolation was performed as previously described (http://www.bio-protocol.org/e906). Briefly, EDTA isolated colonocytes were washed in ice-cold PBS and homogenized by passage through a syringe 20 G needle in breaking buffer (250 mM sucrose, 10 mM Tris (pH 7.4) and protease inhibitors). The 2 ml homogenate was mixed with 2 ml of 62% sucrose and 41.7 μl of 100 mM EDTA (pH 7.4). The 4 ml homogenate was overlayed with 4.5 ml of 35% sucrose solution and 3.5 ml of 29% sucrose solution. Ultracentrifugation was performed in a Beckman SW 41 Ti rotor, at 288,000 × *g* for 1.5 h at 4 °C. After ultracentrifugation, a milky band that was located at the 35%/29% sucrose interface was collected and was diluted with 3 volumes of PBS prior to pelleting. Pellets were resuspended in CelLytic M buffer (Sigma, C2978) and protease inhibitors.

## Ex vivo analysis of colonic mucus

Distal colonic mucus secretion and processing was assessed by quantifying the ex vivo mucus growth rate as previously described[38]. Colon tissue was chamber mounted, basolateral perfusion was set to 5 ml/h, and chambers were heated to 37 °C. For experiments examining senGC-dependent/independent mucus secretory responses, tissue was apically treated with ultrapure LPS from *Escherichia coli* 0111:B4 (200 μg/ml; Invivogen). The mucus and tissue were observed under a stereomicroscope (Leica MZ12⁵) and the mucus thickness measured using a 5-μm-diameter micropipette attached to a micrometer (Mitotoyo). Mucus thickness measurements were acquired at over a 30–60-min period and mucus growth rates expressed as micrometers per minute.

## 16S Fluorescence in situ hybridization (FISH)

16S rRNA FISH was performed as previously described[39]. Briefly, mid-colonic intestinal tissues were prepared for FISH analysis by fixation in Carnoy's fixative, followed by embedding in paraffin as described. Tissues were sectioned and hybridized to a universal bacterial probe directed against the 16S rRNA gene: GCTGCCTCCCGTAGGAGT. Tissues were visualized using a Zeiss Axio Imager M1 Microscope.

## DNA extraction for 16S rRNA analysis

For isolation of luminal contents from the colon, a 4 cm section of colon was cut open longitudinally and a fecal pellet were extracted and weighed. For analysis of tissue-associated bacteria, the same tissue samples that were used for analysis of luminal contents were washed in ice-cold PBS, and then the whole tissue was weighed. Fecal and tissue DNAs were extracted using the FastDNA Spin Kit (MP Biomedicals 116560-200) following the manufacturer's protocol.

## 16S Q-PCR analysis

16S Q-PCR analysis was performed as previously described[13]. Five hundred nanograms of DNA was amplified from each sample using 0.2 μM universal forward primer 27F (5-AGAGTTTGATCMTGGCTCAG-3) and reverse primer 1492R (5-CGGTTACCTTGTTACGACTT-3), and the HotStarTaq polymerase kit (Qiagen, 203203). Thermocycling conditions were 1 cycle of 95 °C for 5 min; 16 cycles of 94 °C for 1 min, 55 °C for 1 min, 72 °C for 1.5 min; 1 cycle of 72 °C for 10 min. The amplified DNA samples, standards and controls were then analyzed by Q-PCR using the SYBR Green kit (Thermo Fisher, 4309155) with universal 16S forward primer (ACTCCTACGGGAGGCAGCAG) and universal 16S reverse primer (ATTACCGCGGCTGCTGG). The total number of 16S copies was determined using standard curves generated from quantified standard plasmids. The 16S data obtained from samples were normalized to the luminal content or tissue weight. The abundance of SFB was determined by Q-PCR using SFB forward primer (GACGCTGAGGCATGAGAGCA) and SFB reverse primer (GACGGCACGGATTGTTATTC). All primers were purchased from IDT technologies.

## Protein digestion and mass spectrometry

Protein from isolated Golgi apparatus were precipitated in 23% TCA and washed with cold acetone. Proteins from Ileum were isolated from the dried supernatant after tissue lysis in extraction buffer (90% methanol, 1% acetic acid)[40]. Proteins were solubilized in 8 M urea 100 mM Tris pH 8.5 and reduced with 5 mM Tris(2-carboxyethyl) phosphine hydrochloride (Sigma-Aldrich, product C4706) and alkylated with 55 mM 2-Chloroacetamide (Sigma-Aldrich, product 22790). Proteins were digested for 18 h at 37 °C in 2 M urea 100 mM Tris pH 8.5, with 0.5 μg trypsin (Promega, Madison, WI, product V5111). Single phase analysis was performed using a QExactive HF mass spectrometer (Thermo Scientific).

For Golgi apparatus enrichment, protein and peptide identification were done with MSFragger (version 17.1)[41] (https://fragpipe. nesvilab.org/) using a mouse protein database downloaded from UniProt (uniprot.org) (one sequence per gene 9/13/2022, 21984 entries), with common contaminants and reversed sequences added by MSFragger. The search space included all fully tryptic peptide candidates with a fixed modification of 57.021464 on C, variable modification of 15.9979 on M and 42.0106 on the N-terminus. MS1 quantification was done with Total Intensity and no match between runs. Protein intensity values were combined for replicates.

For proteins extracted from ileum, protein and peptide identification were done with ProLucid (version1.4)[42] and DTASelect (v2.1.12)[43] using a mouse protein database downloaded from UniProt (uniprot.org) (one sequence per gene 9/13/2022, 21,984 entries), with common contaminants and reversed sequences added by IP2 (Integrated Proteomics Pipeline (version 6.7.1) at The Scripps Research Institute (goldfish.scripps.edu). The search space included all peptide candidates with a fixed modification of 57.02146 on C. Protein spectral counts were combined for replicates.

## Data reproducibility and statistical analysis

All strains were generated and maintained on the same pure inbred background (C57BL/6J); experimental assessment of variance was not performed. No data was excluded. The investigator were not blinded to genotypes or group allocations during any experiment. Comparisons of differences were between two unpaired experimental groups in all cases. An unpaired $t$-test (Student's $t$ test) is appropriate and was used for such comparisons. One-way or two-way ANOVA with post hoc Tukey test was applied to experiments with three or more groups. The phenotype of mice (C57BL/6J) and primary cells of these mice is expected to follow a normal distribution. The statistical significance of differences between experimental groups was determined with GraphPad Prism 7 software and the Student's $t$ test (unpaired, two-tailed). A $P$ value of less than 0.05 was considered statistically significant. No pre-specified effect size was assumed, and in general four mice or more for each genotype or condition were used in experiments; this sample size was sufficient to demonstrate statistically significant differences in comparisons between two unpaired experimental groups by an unpaired $t$-test.

## Key reagents

Please see Supplementary Data 4.

## Reporting summary

Further information on research design is available in the Nature Portfolio Reporting Summary linked to this article.

## Data availability

All raw proteomic data has been deposited at the ProteomeXchange with source code PXD038751, PXD038895. RNA sequencing data is available with reference GSE224516. Source data are provided with this paper.

## Material availability

Further information and requests for resources and reagents should be directed to and will be fulfilled by the Lead Contact, E.E.T. (emre.turer@utsouthwestern.edu).

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

## Acknowledgements

This work was supported by the following grants: NIH R01 DK119360 (E.E.T., R.S.), NIH R03 DK125631 (E.E.T.), NIH K08-DK-123316 (J.N.), T32DK007745 (A.F.), NIH R01-AI125581 (B.B.), U19-AI100627 (B.B.), NIH P41GM103533 (J.Y.), and the Burroughs Wellcome Fund CAMS (J.N.). Glycomics analysis was performed at the Complex Carbohydrate Research Center and was supported in part by the National Institutes of Health (NIH)-funded R24 grant R24GM137782 to (P.A.). E.E.L.N was funded by Svenska Sällskapet för Medicinsk forskning, SSMF (PD20-0168). We would like to thank the Helmsley IBD Exomes Program and the groups that provided exome variant data for comparison. A full list of contributing groups can be found at http://ibd.broadinstitute.org/about. We would also like to thank Andrew Lemoff and the UTSW Mass Spectrometry core for their assistance.

## Author contributions

Conceptualization, E.E.T. and R.S.; Methodology, E.E.T., R.S. and G.B.; Investigation, R.S., W.M., A.M.F., G.B., E.N.G., E.E.L.N, J.C., L.A., S.F., X.L., J.S., J.J.M. and J.N.; Visualization, R.S. and G.B. Writing—original draft, E.E.T., R.S. and E.M.Y.M.; Writing—review & editing, E.E.T., E.N.G., B.B., J.N., and E.M.Y.M.; Funding acquisition, E.E.T., B.B.; Resources, G.B., B.B., J.Y., P.A., E.E.T. and B.B.; Supervision, E.E.T.

## Competing interests

The authors declare no competing interests
