## [Peer Review File · Nature Communications]

Trans-Golgi protein TVP23B regulates host-microbe interactions via Paneth cell homeostasis and Goblet cell glycosylationREVIEWER COMMENTS

Reviewer #1 (Remarks to the Author):

This paper reports the application of a large scale forward genetic screen to identify the Golgi protein TVP23B as being required for normal glycosylation in the intestine of mice, with defects in TVP23B resulting in a defective mucosal layer and increased susceptibility to chemically induced and infection colitis. The TVP23 orthologue in yeast has been shown to form a complex with another small membrane protein of unknown function, and the authors show that mutations in the mouse orthologue of YIPF6 cause similar defects to loss of TVP23, and show that the two proteins interact. They examine Golgi-enriched fractions from both mutants and find that they share the loss of a subset of glycosyltransferases including one known to be required for the synthesis of the major mucin MUC2, thus providing a plausible explanation for the phenotypes observed.

Overall, this is a very high quality study based on an unbiased forward screen. The paper contains a prodigious amount of data which is carefully quantified. Despite the large amount of data, the papers is clearly and succinctly written and I found it easy to follow. The findings highlight the importance of mucins and the cells that make them for protecting the intestine and preventing colitis, and they also highlight the importance of the TVP and YIPF families in Golgi function, two sets of proteins that are very well conserved in evolution but not well characterised. There are a few relatively minor technical points that need to be addressed which are listed below, but apart from these, I am happy to recommend publication of this very impressive study.

Issues to address:

- a) For the naïve reader, the authors should explain the amina propria and muscularis layers of the intestine.
- b) The electronmicrographs in Figures 3 and 4 should be shown as much larger panels to make them easier to see.
- c) In Figure 7B there did not seem to be a band corresponding to FLAG-mCherry control in the "input - IB FLAG" blots, but one should be there. Can the authors check they have the right figure, or provide an explanation for this.
- d) Figure 7D and 7E. Why is the scatter of the data not centred around a fraction of 1.0 as I think it is supposed to be a ratio of mutant to wild type? Also, why is the Y axis labelled "mutant/total" which does not seem to make sense to me?
- e) The authors show that B3GNT6 is lost from the Golgi enriched fractions of mice lacking both Yip6 and Tvp23b (Figure 7G and H). They also state that the levels of B3gnt6 are unchanged in the total cell lysate, but there is no band in the lysate fraction corresponding to the smeary band that is missing from the Golgi in the mutants and so this conclusion does not seem valid. In addition, the B3GNT6 band in the Golgi fraction appears to be of a slightly larger molecular weight in G rather than H.
- f) It would be useful to check by blotting some other Golgi glycosyltransferases in addition to B3GNT6, especially those which are not altered by the mutants.
- g) Supplemental Figure S5 needs size markers for the various gels, and there is typo in the figure legend ("explements").

Reviewer #2 (Remarks to the Author):

The study's central and novel finding is the complementary role of two interacting Golgi proteins, TVP23B and YIP6, on core 3 glycanation of colonic goblet cell mucin, a key protective element in mucosal integrity. It also uncovers impact of TVP23B on Paneth cell protective functionality with

respect to anti-microbial vesicle formation.

Together, these provide new and sophisticated insights linking the effete cell biology of select proteins in the Golgi to the functionality of two protective cellular sentinels of the mucosa. These insights have immediate impact on genetics and disease subtypes of intestinal inflammatory diseases (IBD for sure, but likely other classes of inflammatory conditions spanning the intestine). And, it provides clues to test whether the biology of these molecules (or their related protein isoforms) may play an analogous role in other epithelial compartments or organs.

The experimental work has multiple state-of-art dimensions that strongly support the relationships between TVP23B and/or YIP6 and the observed phenotypes. However, as all the phenotypic work has been assessed *in vivo*, it leaves upon this question: might the molecular compositional deficiencies (notably mucin abundance) be in part or largely secondary to microbiome changes? Certain luminal microbiota proteolytically degrade mucin (especially with deficient glycanation), and others are important in signaling (directly or indirectly) mucin vacuole production and secretion. Since there is a strong Paneth cell phenotype in mice with hypo functional TVP23B, the resultant changes in microbial control and composition could contribute to the observation of reduced mucin vacuole formation and/or accelerated secretion (hence a net reduction of intracellular vacuole contents), and/or accelerated luminal goblet product degradation. The Golgi proteome experiments and concordant deficiencies of glycosylation enzymes in TVP23B and YIP6 is solid evidence that there is a primary glycosylation defect contributing to the phenotype. However, I think targeted study of enteroids in these mutants would provide independent evidence for the scale of goblet vacuole formation deficits, independent of microbiome contribution.

Reviewer #3 (Remarks to the Author):

Using a forward genetic screen (ENU mutagenesis in mice), Song and colleagues identified a mutation in TVP23B to give susceptibility to chemically induced colitis. This was further confirmed by generating a CRISPR TVP23B mutant mouse, which recapitulated susceptibility to DSS colitis, and the authors expanded their findings to *C. rodentium* infection. *Tvp23R* expression and function seem to be restricted to the epithelium (SI and colon), and the loss of function experiment using the KO mice showed dramatic effects in Paneth cell numbers. Although goblet cell numbers seem to be similar between the *Tvp23b*^{-/-} and control mouse, the KO showed defects in mucus production as seen by decreased Alcian blue, mucus thickness decreased epithelium-microbiota gap and higher penetrability of beads. Finally, the authors showed that TVP23B localize at the Golgi compartment, binds YIPF6, and mediated glycosyltransferase functions, which explains the decrease in mucus glycosylation observed in the TVP23B-deficient animals.

This paper is of high interest as it goes from a mutagenesis screening to a novel gene function in colitis. Moreover, it provides an interesting potential mechanism of IBD pathogenesis, involving mucus glycosylation, which is gaining attention in the intestinal barrier field. However, some issues need attention prior to publication

Major concerns

1. The authors performed a forward genetic screen to identify mutations in TVP23B to give susceptibility to DSS-induced colitis. If I understood correctly, they tested 2039 different mice to DSS, which is quite impressive. How many pedigrees show the phenotype? A better description of the overall result of the screening is needed to give justice to such an impressive experiment. A scheme showing the experiment and main results would be useful.
2. In Figure 2c, the authors should show the reconstitution efficiency (frequency of donor cells). Innate lymphoid cells (ILCs) play an important role in DSS- and *C.rodentium*-induced colitis and are radioresistant, which makes them hard to replace by donor BM. The authors should analyze if ILC expresses *Tvp23b* (maybe using single-cell RNA-seq publicly available datasets?).
3. In figure 3 the authors showed a defect in Paneth cell numbers and antimicrobial expression of the ileum from *Tvp23b*^{-/-} compared to WT. This correlates with the decrease of the demilitarized zone (DMZ) observed in Fig 5. This phenotype has been associated with segmented filamentous

bacteria (SFB) levels (PMID: 29139475), which control Th17 (PMID: 19836068), which might then explain intestinal pathology under DSS and *C. rodentium* infection. Therefore, the authors should analyze the SFB levels in the adjacent mucosa and Th17 numbers in SI and colon, which might explain the observed phenotype.

4. In line with the previous comment, mucus has been described to promote tolerance, therefore decrease in mucus may result in the breakdown of tolerogenic mechanisms. Thus, it would be informative to analyze Treg numbers and DC tolerogenic capacity (RA-producing capacity).

5. Figure 5D nicely show the decrease in the so-called demilitarized zone (DMZ) in *Tvp23b*^{-/-} mice compared to controls. However, this needs to be quantified, as the DMZ is variable in the same mouse.

6. Figure S1, performs an immunophenotyping analysis on blood. However, immune responses that are relevant for specific phenotypes are usually located in the target tissue, in this case, the SI and/or colon. I find this figure not relevant and even misleading, and I encourage the authors to perform immunophenotyping in the relevant tissue (SI and/or colon). Moreover, B220 is not enough to define B cells as pDC can also express B220 and some B cells do not. A gating strategy needs to be shown to be able to evaluate the data. Consider, not all macrophages in the gut express F4/80. CD64 is a better marker (PMID: 22936024)

7. Figure S3B-C indicates that 5 mice/genotype has been used however is missing information regarding cages and dams, which enables a better interpretation of the data.

Minor issues

1. Figure 1a shows statistics between REF and VAR, however, the VAR group has only two data points, and therefore is not possible to run statistics using that group. Please correct.

2. Figure supp 1a, has a typo in Peripheral blood (Y axis)

3. Since Nature communications are for broad readers, I recommend explaining what core 1-4 (O-glycans) means.

4. Reference 29 may be outdated (40 years old). I assume the field has moved dramatically considering new technologies to measure proteins.

Response to Referees

We would like to thank the reviewers for their thorough evaluation of our manuscript. We found the comments and critiques were overall very encouraging and helpful in improving the quality of our first study on Tvp23b. We have added additional data including additional glycosyl transferase levels, measurements of organoid vacuoles, quantification of inner mucus layer on 16S FISH micrographs, further immunophenotyping as well as the addition of an epithelial specific conditional targeting. We believe that we have addressed all the major and minor concerns that were raised in the initial review.

Our point by point responses are written in blue.
All additions to the text are in highlighted in yellow.

Reviewer #1 (Remarks to the Author):

This paper reports the application of a large scale forward genetic screen to identify the Golgi protein TVP23B as being required for normal glycosylation in the intestine of mice, with defects in TVP23B resulting in a defective mucosal layer and increased susceptibility to chemically induced and infection colitis. The TVP23 orthologue in yeast has been shown to form a complex with another small membrane protein of unknown function, and the authors show that mutations in the mouse orthologue of YIPF6 cause similar defects to loss of TVP23, and show that the two proteins interact. They examine Golgi-enriched fractions from both mutants and find that they share the loss of a subset of glycosyltransferases including one known to be required for the synthesis of the major mucin MUC2, thus providing a plausible explanation for the phenotypes observed.

Overall, this is a very high quality study based on an unbiased forward screen. The paper contains a prodigious amount of data which is carefully quantified. Despite the large amount of data, the papers is clearly and succinctly written and I found it easy to follow. The findings highlight the importance of mucins and the cells that make them for protecting the intestine and preventing colitis, and they also highlight the importance of the TVP and YIPF families in Golgi function, two sets of proteins that are very well conserved in evolution but not well characterised. There are a few relatively minor technical points that need to be addressed which are listed below, but apart from these, I am happy to recommend publication of this very impressive study.

We appreciate the helpful comments and critiques of the reviewer. We believe that we have addressed your concerns. Our point by point answers are as follows.

Minor Concerns:

a) For the naïve reader, the authors should explain the lamina propria and muscularis layers of the intestine.

We have better defined these terms in the text.

No staining was observed in the underlying lamina propria, which is rich with immune cells, or smooth muscle containing muscularis layers of the intestines of the epitope tagged mice.

b) The electron micrographs in Figures 3 and 4 should be shown as much larger panels to make them easier to see.

We have increased the size of these micrographs in both of the figures.

c) In Figure 7B there did not seem to be a band corresponding to FLAG-mCherry control in the “input – IB FLAG” blots, but one should be there. Can the authors check they have the right figure, or provide an explanation for this.

The FLAG-mCHERRY has a P2A cleavage site between the FLAG and mCHERRY, so the mCHERRY is not tagged by the FLAG epitope. We have updated the figure to reflect this point.

d) Figure 7D and 7E. Why is the scatter of the data not centered around a fraction of 1.0 as I think it is supposed to be a ratio of mutant to wild type? Also, why is the Y axis labelled “mutant/total” which does not seem to make sense to me?

The mass spec data is normalized to the total counts (total= mutant + wildtype) and expressed as a fraction of total (mutant/total). This is done to have all data points between 0 and 1, rather than 0 to infinity. This method of depicting the data is done so that the figure shows both enriched and depleted proteins in the same manner without biasing in either direction.

e) The authors show that B3GNT6 is lost from the Golgi enriched fractions of mice lacking both Yipf6 and Tvp23b (Figure 7G and H). They also state that the levels of B3gnt6 are unchanged in the total cell lysate, but there is no band in the lysate fraction corresponding to the smeary band that is missing from the Golgi in the mutants and so this conclusion does not seem valid. In addition, the B3GNT6 band in the Golgi fraction appears to be of a slightly larger molecular weight in G rather than H.

Unfortunately, we do not have B3gnt6 deficient mice as a control for antibody specificity. The antibody has previously been published to detect B3GNT6 protein by immunoblot^{1, 2}. We have interpreted the upper smear to be a glycosylated form of B3GNT6 as many Golgi resident proteins will accrue this modification. Two glycosylation sites have been identified on B3GNT6 by mass spectrometry³. The lower band in the WCL corresponds to the native molecular weight of the protein and is likely unmodified. We have updated the text of the manuscript to reflect that the modified and likely glycosylated form is lost.

The Golgi fraction displayed a higher molecular weight, presumably glycosylated, B3GNT6, which was lost in mutant animals (Figure 7G, H). This protein loss is in contrast to another Golgi localized glycosyltransferase, Galnt4, which is unchanged in the mutant (Figure S8).

f) It would be useful to check by blotting some other Golgi glycosyltransferases in addition to B3GNT6, especially those which are not altered by the mutants.

We have added additional immunoblots for Galnt4, which is added to the supplemental figures (Figure S8). See below.

Figure S8

g) Supplemental Figure S5 needs size markers for the various gels, and there is typo in the figure legend (“explements”).

We have added the size markers to Figure S7 (Formerly Figure S5) and fixed the typo. See below.

Figure S7

Reviewer 2:

The study's central and novel finding is the complementary role of two interacting Golgi proteins, TVP23B and YIP6, on core 3 glycanation of colonic goblet cell mucin, a key protective element in mucosal integrity. It also uncovers impact of TVP23B on Paneth cell protective functionality with respect to anti-microbial vesicle formation.

Together, these provide new and sophisticated insights linking the effete cell biology of select proteins in the Golgi to the functionality of two protective cellular sentinels of the mucosa. These insights have immediate impact on genetics and disease subtypes of intestinal inflammatory diseases (IBD for sure, but likely other classes of inflammatory conditions spanning the intestine).

And, it provides clues to test whether the biology of these molecules (or their related protein isoforms) may play an analogous role in other epithelial compartments or organs.

The experimental work has multiple state-of-art dimensions that strongly support the relationships between TVP23B and/or YIP6 and the observed phenotypes. However, as all the phenotypic work has been assessed *in vivo*, it leaves upon this question: might the molecular compositional deficiencies (notably mucin abundance) be in part or largely secondary to microbiome changes? Certain luminal microbiota proteolytically degrade mucin (especially with deficient glycanation), and others are important in signaling (directly or indirectly) mucin vacuole production and secretion. Since there is a strong Paneth cell phenotype in mice with hypo functional TVP23B, the resultant changes in microbial control and composition could contribute to the observation of reduced mucin vacuole formation and/or accelerated secretion (hence a net reduction of intracellular vacuole contents), and/or accelerated luminal goblet product degradation.

The Golgi proteome experiments and concordant deficiencies of glycosylation enzymes in TVP23B and YIP6 is solid evidence that there is a primary glycosylation defect contributing to the phenotype. However, I think targeted study of enteroids in these mutants would provide independent evidence for the scale of goblet vacuole formation deficits, independent of microbiome contribution.

We appreciate the reviewer's comments. The use of organoids would help demonstrate epithelial specificity as well as eliminate the microbiota as a confounding factor.

We have generated colonoids from *Tvp23b^{+/+}* and *Tvp23b^{-/-}* mice and differentiated them to the goblet lineage using DAPT and IWP-2. Colonoids were stained with PAS to demonstrate the vacuole defect. We have added representative micrographs and quantitative measurements of vacuole size as a subpanel to Figure 4C-D. Compartment size was quantified by using ImageJ to measure PAS+ vacuoles in organoids derived from 3 different mice of each genotype. See below.

A similar phenotype was found in goblet cell differentiated organoid cultures with smaller average PAS staining vacuole size (7.9 μm vs 5.3 μm) in the *Tvp23b^{-/-}* cells thus demonstrating that the phenotype is epithelial intrinsic and not dependent on the microbiome (Figure 4C, D).

Reviewer 3:

Using a forward genetic screen (ENU mutagenesis in mice), Song and colleagues identified a mutation in TVP23B to give susceptibility to chemically induced colitis. This was further confirmed by generating a CRISPR TVP23B mutant mouse, which recapitulated susceptibility to DSS colitis, and the authors expanded their findings to *C. rodentium* infection. Tvp23R expression and function seem to be restricted to the epithelium (SI and colon), and the loss of function experiment using the KO mice showed dramatic effects in Paneth cell numbers. Although goblet cell numbers seem to be similar between the Tvp23b^{-/-} and control mouse, the KO showed defects in mucus production as seen by decreased Alcian blue, mucus thickness decreased epithelium-microbiota gap and higher penetrability of beads. Finally, the authors showed that TVP23B localize at the Golgi compartment, binds YIPF6, and mediated glycosyltransferase functions, which explains the decrease in mucus glycosylation observed in the TVP23B-deficient animals.

This paper is of high interest as it goes from a mutagenesis screening to a novel gene function in colitis. Moreover, it provides an interesting potential mechanism of IBD pathogenesis, involving mucus glycosylation, which is gaining attention in the intestinal barrier field. However, some issues need attention prior to publication

We appreciate the care and diligence of reviewer comments and believe we have addressed the concerns. Our point-by-point responses are as follows:

1. The authors performed a forward genetic screen to identify mutations in TVP23B to give susceptibility to DSS-induced colitis. If I understood correctly, they tested 2039 different mice to DSS, which is quite impressive. How many pedigrees show the phenotype? A better description of the overall result of the screening is needed to give justice to such an impressive experiment. A scheme showing the experiment and main results would be useful.

In actuality, 55,867 G3 mice were tested from 2039 different G1 pedigrees. This level of mutation corresponds to 26% genomic damage tested at least three times⁴. Only one pedigree had a mutation in TVP23B resulting in phenovariance. There were 306 mice that lost 20% of starting body weight, which was ~0.5% of the total mice tested and were found in 174 pedigrees. Interim results from this screen have been previously published^{5, 6, 7}. We have added additional text and references describing the screen as well as a supplemental figure summarizing the screening scheme and the results.

“We subjected 55,867 third generation (G3) mice from 2,039 pedigrees to DSS in their drinking water, and body weights of the mice were recorded after 7 days of DSS treatment^{5, 6, 7}. Among the 55,86 G3 mice tested, a total of 306 mice (0.5% of total) displayed weight loss of 20% by day 7 of DSS treatment (Figure S1).”

Editorial Note: Figure S1A was generated with BioRender.com.

Figure S1

2. In Figure 2c, the authors should show the reconstitution efficiency (frequency of donor cells).

The reconstitution efficiency was assessed in peripheral blood and ranged from 94-97% irrespective of genotype. Unfortunately, this was not done in the lamina propria cells, which is more relevant. Shown is a representative reconstitution in the peripheral blood.

We have added the following to the text:

The reconstitution efficiency of donor cells ranged from 94-97% in the peripheral blood.

Innate lymphoid cells (ILCs) play an important role in DSS- and C. rodentium-induced colitis and are radioresistant, which makes them hard to replace by donor BM. The authors should analyze if ILC expresses Tvp23b (maybe using single-cell RNA-seq publicly available datasets?).

Using available scRNAseq dataset of the murine Lamina Propria, it does not appear that Tvp23b is substantially expressed in ILCs (Fractions #3, 11, 12, 15)⁸. The major cells expressing TVP23B in this dataset were epithelial cells (#35), endothelial cells (#25), fibroblasts (#33), and doublets (#29 and 41). Likely any expression in ILCs is much smaller compared to these non-hematopoietic cells.

The Immgen database does show a modest level expression in ILCs, although no intestinal epithelial cells were present in this dataset for comparison.

To address the question of whether this is truly epithelial intrinsic or due to poor reconstitution of a rare cell type, we have generated a conditional knockout allele (*Tvp23b^{Fl/Fl}*) and examined Villin-CRE animals, which have epithelial specific deletion. These animals, similar to the global knock out, having increased sensitivity to DSS colitis (Figure 2F-H). While not eliminating additional

non-epithelial cell contributions, this data does demonstrate that expression is required in the epithelium to prevent colitis. To eliminate any contributions from ILCs or other hematopoietic cells would require additional CRE strains and be outside the scope of the current study.

Figure 2

Additionally, the recently added organoid data is consistent with this epithelial intrinsic defect and not dependent on ILCs nor microbiota. Please see answer to reviewer #2.

3. In figure 3 the authors showed a defect in Paneth cell numbers and antimicrobial expression of the ileum from *Tvp23b^{-/-}* compared to WT. This correlates with the decrease of the demilitarized zone (DMZ) observed in Fig 5. This phenotype has been associated with segmented filamentous bacteria (SFB) levels (PMID: 29139475), which control Th17 (PMID: 19836068), which might then explain intestinal pathology under DSS and *C. rodentium* infection. Therefore, the authors should analyze the SFB levels in the adjacent mucosa and Th17 numbers in SI and colon, which might explain the observed phenotype.

We have performed qPCR for SFB and there is no statistically significant increase in SFB levels in knockout samples compared to littermate controls. This has been added to the supplemental figure.

The control and levels of segmented filamentous bacteria (SFB) are a key contributor to intestinal inflammatory activation⁹. In *Tvp23b^{-/-}*, however, no significant increase in SFB was observed in tissue associated bacteria (Figure S5D).

We have also analyzed the Th17 cells in the small intestine and colon as shown below. There is no increase in Th17 cells in small intestine TVP23B-deficient animals. In the colons, there is a slight decrease in Th17 cells in knock out animals, which is consistent *Vamp8*^{-/-} mice, which also have a mucus secretion defect¹⁰. We have added this to supplemental figure S3.

4. In line with the previous comment, mucus has been described to promote tolerance, therefore decrease in mucus may result in the breakdown of tolerogenic mechanisms. Thus, it would be informative to analyze Treg numbers and DC tolerogenic capacity (RA-producing capacity).

We have further phenotyped by FACS the intestinal immune compartment of the *Tvp23b*^{-/-} mice including the regulatory T cell and regulatory DC compartments. We have also measured the RA producing capacity of the lamina propria DCs (CD45+, MHC II+ CD11c+ CD64-). Shown below are the results. There is no difference in the frequency of FOXP3+ cells (CD45+ TCRb+ CD4+) in the small intestine or colon. These are added to the supplemental figure S3.

5. Figure 5D nicely show the decrease in the so-called demilitarized zone (DMZ) in *Tvp23b*^{-/-} mice compared to controls. However, this needs to be quantified, as the DMZ is variable in the same mouse.

We agree that the DMZ or inner mucus layer is variable, which is why we did not rely simply on 16S FISH, but also the bead penetration assay (Figure 5B, C) as well as the overall mucus thickness and growth (Figure 4E-G). We have added further quantitation of the 16S FISH averaging the distance across 20 points per mouse (n=4 per genotype). This quantitation has been added to figure 5E.

E
6. Figure S1, performs an immunophenotyping analysis on blood. However, immune responses that are relevant for specific phenotypes are usually located in the target tissue, in this case, the SI and/or colon. I find this figure not relevant and even misleading, and I encourage the authors to perform immunophenotyping in the relevant tissue (SI and/or colon). Moreover, B220 is not enough to define B cells as pDC can also express B220 and some B cells do not. A gating strategy needs to be shown to be able to evaluate the data. Consider, not all macrophages in the gut express F4/80. CD64 is a better marker (PMID: 22936024)

We thank the reviewer for pointing out the oversight. In the peripheral blood, we have stained B cells for both B220 and CD19 and have corrected this in the figure legend. We also agree that this examining tissue specific immune homeostasis is more relevant. As above we have added additional immunophenotyping including FACS gating schemes in supplemental figures (S2 and S3).

Figure S2- gating strategy

Figure S3- gating strategy

Figure S3- colon immune homeostasis

7. Figure S3B-C indicates that 5 mice/genotype has been used however is missing information regarding cages and dams, which enables a better interpretation of the data.

We believe the microbial composition shift to be relatively minor compared to the increased penetrability of microbes due to defective mucus. We have added data regarding dams and cages to the figure legend. The mice are derived from 4 different dams and each experimental mouse had at least one corresponding cage/littermate.

No.	Ear ID	DOB	Genotype	Sex	Cage	dam
1	B14076	8/21/20	Het	M	# 1	S96421
2	F15004	5/10/21	Het	F	# 2	B16330
3	F15005	5/10/21	Het	F	# 2	B16330
4	B20056	1/1/21	Het	M	# 3	B15214
5	F15091	6/8/21	Het	M	# 4	B22163
6	B14075	8/21/20	Homo	M	# 1	S96421
7	F15002	5/10/21	Homo	F	# 2	B16330
8	F15006	5/10/21	Homo	F	# 2	B16330
9	B20054	1/1/21	Homo	M	# 3	B15214
10	F15092	6/8/21	Homo	M	# 4	B22163

Minor issues

1. Figure 1a shows statistics between REF and VAR, however, the VAR group has only two data points, and therefore is not possible to run statistics using that group. Please correct.

We have corrected this in Figure 1A.

2. Figure supp 1a, has a typo in Peripheral blood (Y axis)

We have corrected the typo.

3. Since Nature communications are for broad readers, I recommend explaining what core 1-4 (O-glycans) means.

We have added a supplemental figure depicting the different carbohydrate linkages.

“O-glycosylation cores 1-4 are determined by the initial sugar moiety added to the protein (Figure S6A).”

4. Reference 29 may be outdated (40 years old). I assume the field has moved dramatically considering new technologies to measure proteins.

We agree this field has moved dramatically over the last 40 years. We have added additional references including several reviews addressing the topic.

References:

1. Wu H, *et al.* G-quadruplex-enhanced circular single-stranded DNA (G4-CSSD) adsorption of miRNA to inhibit colon cancer progression. *Cancer Med*, (2023).
2. Ye J, *et al.* Core 3 mucin-type O-glycan restoration in colorectal cancer cells promotes MUC1/p53/miR-200c-dependent epithelial identity. *Oncogene* **36**, 6391-6407 (2017).
3. UNIPROT-B3GN6.

4. Wang T, *et al.* Probability of phenotypically detectable protein damage by ENU-induced mutations in the Mutagenetix database. *Nat Commun* **9**, 441 (2018).
5. McAlpine W, *et al.* Excessive endosomal TLR signaling causes inflammatory disease in mice with defective SMCR8-WDR41-C9ORF72 complex function. *Proc Natl Acad Sci U S A* **115**, E11523-E11531 (2018).
6. Turer E, *et al.* Creatine maintains intestinal homeostasis and protects against colitis. *Proc Natl Acad Sci U S A* **114**, E1273-E1281 (2017).
7. Wang KW, *et al.* Enhanced susceptibility to chemically induced colitis caused by excessive endosomal TLR signaling in LRBA-deficient mice. *Proc Natl Acad Sci U S A* **116**, 11380-11389 (2019).
8. Xu H, *et al.* Transcriptional Atlas of Intestinal Immune Cells Reveals that Neuropeptide alpha-CGRP Modulates Group 2 Innate Lymphoid Cell Responses. *Immunity* **51**, 696-708 e699 (2019).
9. Ivanov, II, *et al.* Induction of intestinal Th17 cells by segmented filamentous bacteria. *Cell* **139**, 485-498 (2009).
10. Cornick S, Kumar M, Moreau F, Gaisano H, Chadee K. VAMP8-mediated MUC2 mucin exocytosis from colonic goblet cells maintains innate intestinal homeostasis. *Nat Commun* **10**, 4306 (2019).

REVIEWERS' COMMENTS

Reviewer #1 (Remarks to the Author):

The authors have engaged effectively with my comments and made the additions and corrections that I suggested. I am thus happy to recommend acceptance of this interesting and extensive study.

Reviewer #2 (Remarks to the Author):

The revision fully addresses my set of critiques. In my judgment, the same is true for the other sets of critiques.

Reviewer #3 (Remarks to the Author):

After careful consideration of the authors' revisions, I am pleased to note that they have satisfactorily addressed all of the concerns and suggestions raised in my previous comments. The manuscript has been significantly improved, and I am now confident in its clarity and scientific rigor. I appreciate the authors' efforts in addressing the comments and their commitment to enhancing the quality of their work.

Minor comments:

New references need to be updated in the revised main text

RESPONSE TO REVIEWERS' COMMENTS

We would like to thank all three reviewers for their work. We feel the manuscript has improved significantly since the first submission.

Reviewer #1 (Remarks to the Author):

The authors have engaged effectively with my comments and made the additions and corrections that I suggested. I am thus happy to recommend acceptance of this interesting and extensive study.

Answer: We thank the reviewer for their work.

Reviewer #2 (Remarks to the Author):

The revision fully addresses my set of critiques. In my judgment, the same is true for the other sets of critiques.

Answer: We thank the reviewer for their thoughtful comments.

Reviewer #3 (Remarks to the Author):

After careful consideration of the authors' revisions, I am pleased to note that they have satisfactorily addressed all of the concerns and suggestions raised in my previous comments. The manuscript has been significantly improved, and I am now confident in its clarity and scientific rigor. I appreciate the authors' efforts in addressing the comments and their commitment to enhancing the quality of their work.

Minor comments:

New references need to be updated in the revised main text

Answer: We thank the reviewer for their efforts. We have checked all the new references in the main text of the manuscript and made sure the new requested references are present.